# AVATAR: Optimizing LLM Agents for Tool Usage via Contrastive Reasoning

**Shirley Wu**[§], **Shiyu Zhao**[§], **Qian Huang**[§], **Kexin Huang**[§], **Michihiro Yasunaga**[§], **Kaidi Cao**[§]
**Vassilis N. Ioannidis**[†], **Karthik Subbian**[†], **Jure Leskovec**[*§], **James Zou**[*§]
[*]Equal senior authorship.
[§]Department of Computer Science, Stanford University    [†]Amazon

## Abstract

Large language model (LLM) agents have demonstrated impressive capabilities in utilizing external tools and knowledge to boost accuracy and reduce hallucinations. However, developing prompting techniques that enable LLM agents to effectively use these tools and knowledge remains a heuristic and labor-intensive task. Here, we introduce AVATAR, a novel and automated framework that optimizes an LLM agent to effectively leverage provided tools, improving performance on a given task. During optimization, we design a comparator module to iteratively deliver insightful and comprehensive prompts to the LLM agent by contrastively reasoning between positive and negative examples sampled from training data. We demonstrate AVATAR on four complex multimodal retrieval datasets featuring textual, visual, and relational information, and three general question-answering (QA) datasets. We find AVATAR consistently outperforms state-of-the-art approaches across all seven tasks, exhibiting strong generalization ability when applied to novel cases and achieving an average relative improvement of 14% on the Hit@1 metric for the retrieval datasets and 13% for the QA datasets. Code and dataset are available at https://github.com/zou-group/avatar.

## 1 Introduction

Autonomous agents powered by large language models (LLMs) offer substantial promise for complex problem-solving [6, 39, 41, 55, 65]. These agents demonstrate remarkable capabilities in reasoning [46, 47, 54, 55] and planning [8, 13, 14, 62]. Additionally, their functionality is extended through the use of external tools that provide access to external or private data and specialized operations, such as APIs for interacting with knowledge bases and search engines. These tools enable agents to perform complex tasks like multi-step problem-solving and retrieving diverse information, which is essential for complex retrieval and question-answering (QA) [13, 21, 26, 33, 38, 40, 48].

Despite the promising capabilities of LLM agents, it remains challenging to engineer effective prompts that guide these agents through a multi-stage process for real-world problem-solving. This process involves (1) decomposing a complex question into an actionable plan with simpler steps, (2) strategically using provided tools to gather relevant information, and, finally, (3) synthesizing intermediate results to produce a coherent and accurate response. Each step requires extensive manual effort and numerous iterations of trial and error to refine the prompts.

Current approaches have primarily focused on directly deploying agents using complex human-designed "mega-prompts" [18, 24, 55], which require lots of manual trial and error. Nevertheless, such hand-engineered mega-prompts may also result in brittle implementations with suboptimal accuracy (see Figure 2 (a)), where the ReAct agent [55] easily produces trivial and misleading answers to

---

Correspondence: {shirwu, jure, jamesz}@cs.stanford.edu

38th Conference on Neural Information Processing Systems (NeurIPS 2024).

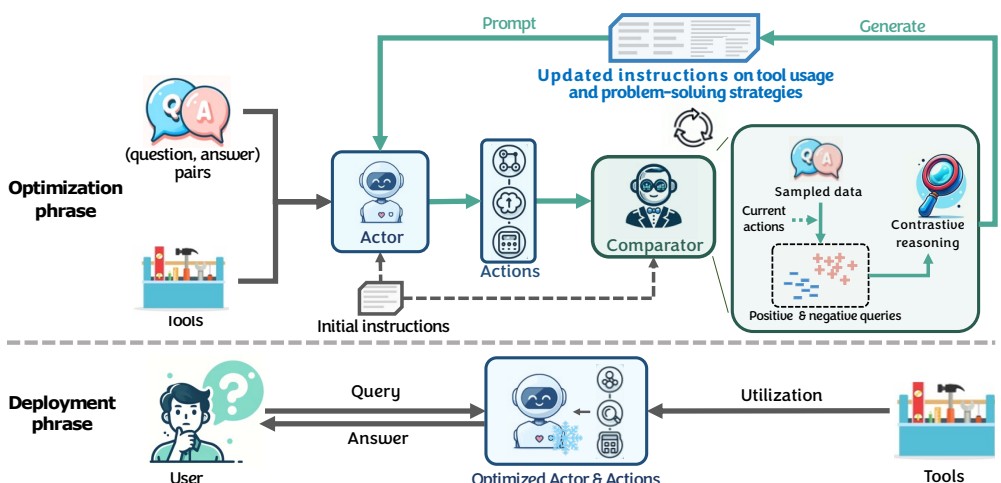

Figure 1: **Overview of AVATAR.** AVATAR consists of a actor LLM and a comparator LLM. (a) During optimization, the actor generates actions to answer queries by leveraging the provided tools. Then, the comparator contrasts a set of well-performing (positive) and poorly-performing (negative) queries, automatically generating holistic prompts to teach the actor more effective retrieval strategies and tool usage (*cf.* Section 4). (b) At deployment, the actor with optimized prompts or actions can be effectively used to answer new queries.

customers' queries about specific products. Furthermore, existing research [4, 5, 45, 50, 56, 60, 64] on employing LLMs as optimizers often fails to adequately refine the complex strategies for enhancing tool integration and usage. This lack of strategic optimization can lead to less effective, non-generalizable agent applications in complex real-world scenarios.

**Present work: AVATAR**. To address these challenges, we introduce AVATAR, an automated framework that optimizes agents for effective tool utilization and excellent task performance. Specifically, we leverage key insights from contrastive reasoning and build a comparator module ("trainer") to generate holistic instructions and prompts (*i.e.,* , computing a robust "gradient") to optimize an actor LLM. We demonstrate our framework on challenging tasks of knowledge base retrieval, which involve complex multi-stage procedures and extensive tool usage, and general QA tasks. Specifically, AVATAR includes two phases:

- **Optimization phase**. The core of our optimization framework (Figure 1) is a comparator LLM that automatically generates holistic prompts to teach a actor LLM to differentiate between effective and ineffective tool usage. The comparator takes positive and negative data samples, where the current agent performs well and poorly, respectively, to identify overall gaps and systematic errors exhibited by the agent. Unlike per-sample instructions, which can easily lead to overfitting on individual data points, by constructing multiple samples as a "batch," the comparator can extract a more robust "gradient" to "backpropagate" to the actor. In other words, the comparator can provide more effective and adaptive prompts through batch-wise contrastive reasoning, helping the agent identify flaws in solving challenging multi-stage problems. Following previous methods [30, 41, 56, 63], we also maintain a memory bank with selected past instructions to prevent the actor LLM from repeating previous mistakes.

- **Deployment phase**. After the optimization phase, the actor with best-performing prompts can be selected for the testing instances. Moreover, in complex retrieval tasks, the iterative optimization through our AVATAR framework updates the actor for more effective and generalizable action sequences, enabling direct generalization to novel user inquiries at deployment. In Figure 2 (b), the optimized actor creates three novel strategies: 1) precise decomposition of problems by extracting multifaceted attributes, 2) effective tool usage through a sophisticated and robust scoring system, and 3) the strategic combination of different scores, determined by learned coefficients, ensuring accurate and comprehensive retrieval.

**Experimental evaluation**. We conduct extensive experiments on four retrieval datasets and three QA datasets. The retrieval tasks are highly complex, involving multimodal data, including textual, visual, and relational information. AVATAR consistently outperforms state-of-the-art methods, showing a

substantial 14% improvement in the Hit@1 metric. Impressively, with only 25 iterations, AVATAR boosts the Hit@1 metric from an initial 5.1% to 28.6% on FLICKR30K-ENTITIES [35] and the Recall@20 metric from 30.3% to 39.3% on STARK-PRIME [49]. For general QA datasets, AVATAR outperforms state-of-the-art methods by 13% on average. These improvements, achieved through iterative updates to the prompts, underscore AVATAR's ability to optimize agents for complex tasks and effective tool usage. Our key contributions are:

- We introduce AVATAR, a novel framework that optimizes an actor for effective tool utilization through a comparator module that automatically generates holistic prompts.
- We demonstrate AVATAR on four complex retrieval tasks and three QA tasks, where it significantly outperforms existing agent methods in terms of task performance and generalization ability.
- We provide a comprehensive analysis of the actor's evolution during optimization, highlighting how comparator automatically provides targeted instructions that improve and generalize the actor.

## 2 Related Work

**LLM Agents**. Recent research has leveraged the remarkable language understanding and reasoning abilities of LLMs [1, 41, 47, 54, 55] to complete downstream tasks. For complex tasks that require enhanced capabilities, previous works have positioned LLMs as agents that can interact with environments [4, 6, 13, 18, 21, 26, 27, 40, 48, 55], leverage external tools [6, 28, 31, 33, 36, 38, 39, 66, 68], and gather experiences [7, 61]. For example, ReAct [55] conducts reasoning and action in an interleaved way, retrieving information from Wikipedia to support reasoning.

**LLM Agents for Retrieval**. Previous research has applied LLM agents to Information Retrieval (IR) systems through pretraining [2, 9, 16, 57], reranking [12, 42], and prompting techniques [11, 18]. In IR systems, the retriever module directly influences the performance of downstream tasks, such as retrieval-augmented generation [20, 29, 30] and knowledge-intensive question answering [34, 52]. For example, EHRAgent [40] is designed for EHR question-answering, capable of retrieving relevant clinical knowledge through a structured tool-use planning process and an interactive coding mechanism. However, these LLM agents usually employ heuristic (zero-shot) prompts or rely on few-shot examples [18, 25, 40, 55] for downstream tasks, which lack more informed guidance on generating effective retrieval strategies and tool-assisted actions.

**Agent Optimization**. In the field of optimizing LLM agents, previous works have modified the parameters of LLM backbones through fine-tuning or instruction tuning to enhance agent capability [3, 15, 19, 23, 32, 33, 37, 43, 51, 58, 59] or generated better prompts through iterative prompt tuning [11, 18, 45, 50, 56]. Recently, Zhang et al. [60] conducted agent training by iteratively updating the agents' functions according to the execution history. However, these methods do not explicitly consider targeted optimization for tool usage or the impact on complex multi-stage tasks. Additionally, enhancing agents' generalization abilities [10, 31, 44], essential for real-world applications, has received less attention. In our work, we focus on automatically generating holistic instructions via a novel contrastive reasoning mechanism, targeting effective tool usage and agents' generalization ability. Compared to fine-tuning approaches, AvaTaR only require a small subset of training data and tool descriptions, making it more adaptable and less computationally intensive.

## 3 Problem Formulation

**Definition 1: Tools**. We define tools or APIs as a set of implemented functions with specified input and output variables. We denote the abstract tool space as $\mathcal{T} = \{f_k : \mathcal{I}_{f_k} \to \mathcal{O}_{f_k} \mid k = 1, 2, \dots\}$, where $f_k$ maps the input $\mathcal{I}_{f_k}$ to the output $\mathcal{O}_{f_k}$. For example, the tools can be APIs used for accessing external knowledge via a search index, an encoder model that generates vector representations from text or image data, or a task-specific classifier that outputs probabilities over a list of classes.

**Definition 2: Agents**. An LLM agent, defined as $\mathcal{A} : \mathcal{P} \to \alpha$, is controlled by verbal prompts to generate a flow of actions needed to complete a task. Here $\alpha$ denotes the action sequence $[\alpha_1, \dots, \alpha_L]$, where each action is defined by a tuple $(f \in \mathcal{T}, i \in \mathcal{I}_f, o \in \mathcal{O}_f)$, consisting of a tool function, specified input(s), and a designated variable that receives the output(s). Each action in the sequence can leverage the outputs generated by previous actions, with the final action $\alpha_L$ rendering the results for the task.

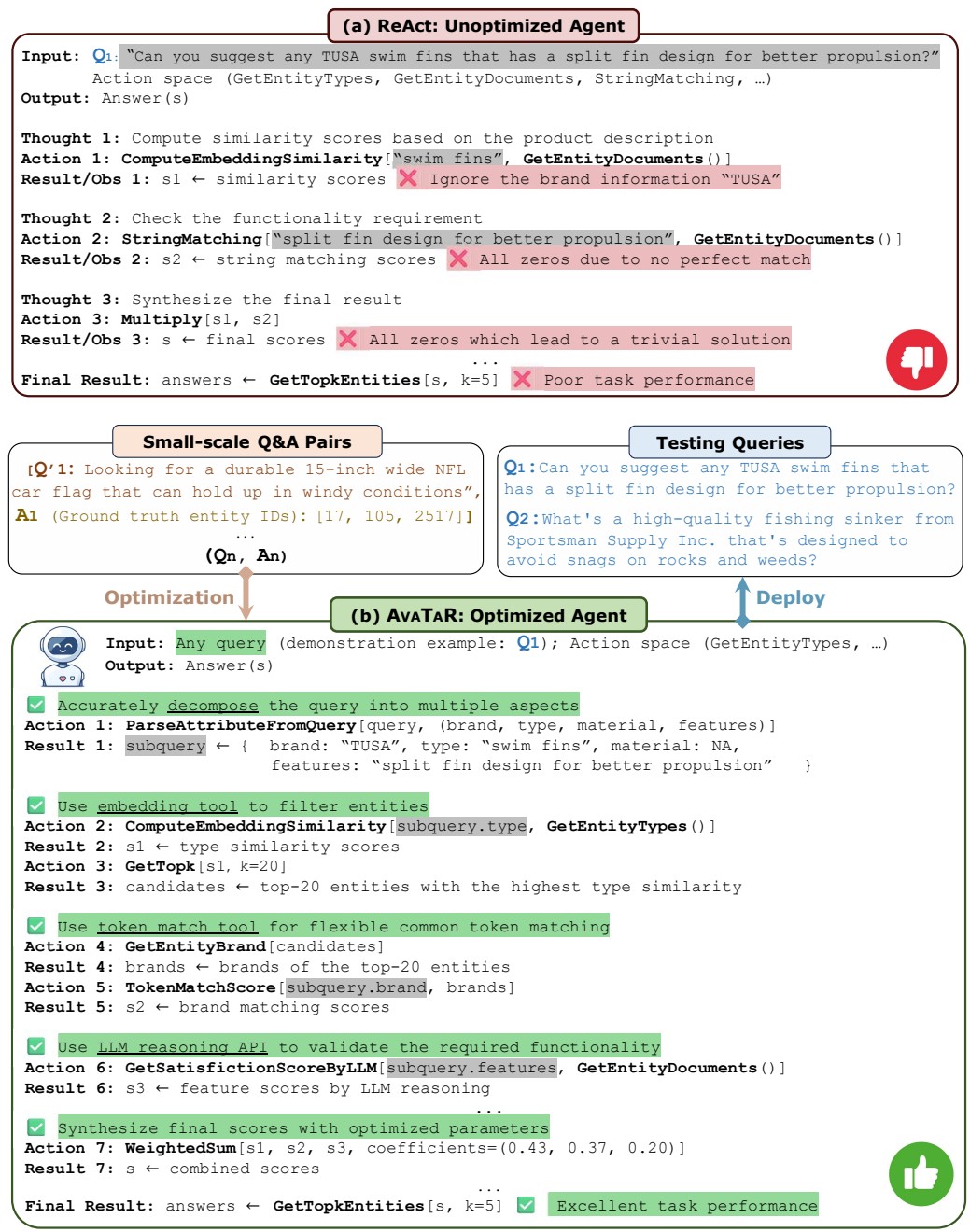

Figure 2: **Comparison between AVATAR and ReAct**. (a) The ReAct agent exhibits incomplete task decomposition and employs suboptimal tool combinations, such as lengthy string matching, leading to poor task performance. (b) AVATAR decomposes the task into multiple steps, such as type filtering and flexible token matching. Moreover, it implements robust tool usage and precise synthesis with learned parameters from the optimization phase to achieve excellent performance on new queries.

**Multi-step problem-solving**. Real-world problems are inherently complex and cannot be effectively addressed through straightforward solutions or simple tool usage alone. Solving real-world problems with LLM agents can be structured into a multi-stage procedure:

- **Decomposition of the problem**: The procedure begins by breaking down a complex question into an actionable plan characterized by simpler steps. This decomposition is crucial for setting clear objectives and facilitating focused problem-solving.

- **Tool-assisted subproblem solving**: In the subsequent phase, agents strategically utilize tools from the established tool space $\mathcal{T}$ to gather solutions for each step. This stage is essential for acquiring

Table 1: **Key differences between AVATAR and prevailing agent methods.** AVATAR demonstrates the ability to: 1) self-improve on specific tasks, 2) retain memory throughout the optimization process, 3) enhance the agent's generalization capability, and 4) autonomously generate holistic, high-quality prompts for better tool usage. Please refer to Section 4 for details.

| | Self-Improvement | Memory | Generalization | Holistic Prompt Generation (on Tool Usage) |
|---|---|---|---|---|
| ReAct [55] | ✗ | ✗ | ✗ | ✗ |
| Self-refine [27] | ✔ | ✗ | ✗ | ✗ |
| Reflexion [41] | ✔ | ✔ | ✗ | ✗ |
| AVATAR (Ours) | ✔ | ✔ | ✔ | ✔ |

the necessary information required to effectively address each subproblem of the decomposed problem.

- **Synthesis and response formulation**: The final stage involves synthesizing the intermediate results to construct a precise response. This synthesis not only combines the data but may also refine the response through trials and adjustments, ensuring the solution's accuracy and relevance.

For example, retrieval tasks are inherently complex and demanding. Given a user query $q$, retrieval tasks aim to identify or generate a ranked list of relevant entities $E$ from the entity space of a knowledge base. Each query is associated with a set of ground truth answers, denoted as $Y$, which are used to compute the quality of the prediction. Specifically, the LLM agent is required to 1) comprehend a user's request, 2) utilize the provided tools to identify and analyze relevant information in the large knowledge space, which may contain multimodal data sources, and finally, 3) integrate all gathered information to reason and generate an accurate response.

## 4 Our Method: Optimizing Agents for Tool-Assisted Multi-Step Tasks

Each step in the multi-stage problem-solving process (described in Section 3) requires effective prompts to identify key flaws and improve task performance. However, refining the agents' prompts demands extensive manual effort and numerous iterations of trial and error.

To address this, we introduce an automated and novel optimization framework, AVATAR, which generates prompts to improve agents' tool usage and task performance. In Table 1, we highlight four critical aspects of our approach compared with prevailing agent frameworks [27, 41, 55]. Here, we introduce the two main LLM components in AVATAR: a actor LLM (Section 4.1) and a comparator LLM (Section 4.2).

### 4.1 Actor Construction and Challenges

**Actor**. The actor agent, as defined in Section 3, is responsible for generating initial actions based on the initial instructions/prompts and adjusting actions according to updated instructions. Specifically, the initial instructions provide details about the task and available tools, where tools can be introduced in programming languages such as Python. During optimization, the prompts further incorporate the previous action sequence and updated instructions to adjust these actions. The actor then generates revised actions, which could include a combination of tool usage through programming language (code generation) along with natural language explanations of how the tools are employed.

**Challenges in multi-step complex tasks**. A common approach to updating instructions utilizes execution results or performance data from a specific instance, often through techniques like self-explanation [4, 27] or self-reflection [41, 56]. However, this approach may not be suitable for complex tasks involving tool usage. Complex multi-step tasks include multiple interacting factors that influence overall performance, such as problem decomposition and tool selection. Consequently, instructions generated for a failed/negative query instance tend to be narrow in scope and may fail to identify flaws across all components of a complex solution. Additionally, while certain tool combinations may be effective for one type of input, their effectiveness can vary across different scenarios, potentially leading to decreased performance when applied to varied cases.

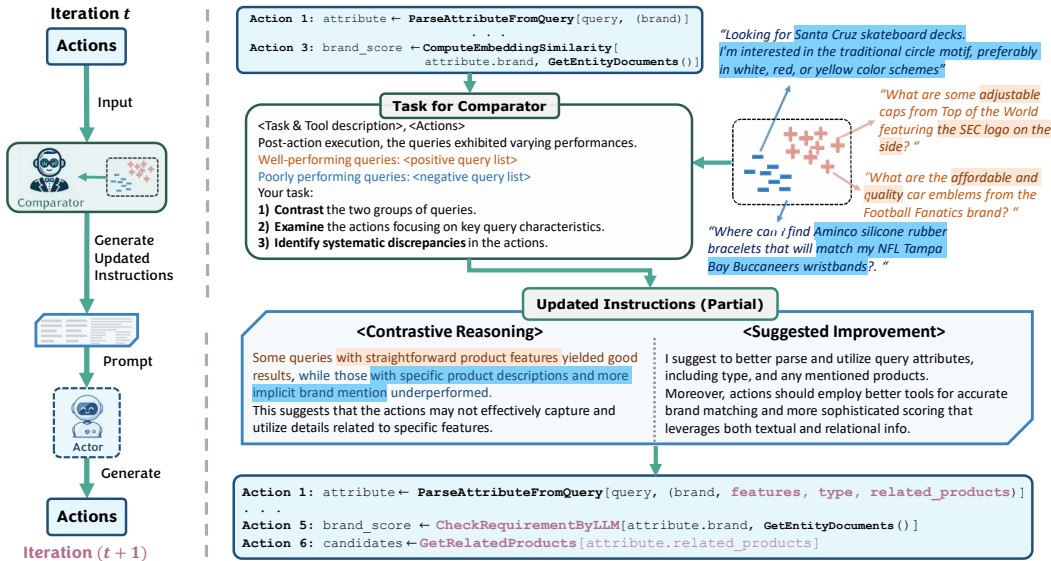

Figure 3: **Demonstration example during optimization.** Best viewed in color. The task of the comparator is to automatically generate instructions based on sampled positive and negative queries. Then comparator provides holistic instructions that guide the actor to improve query decomposition, utilize better tools, and incorporate more comprehensive information.

## 4.2 Automate Holistic Instruction Generation with Comparator

To address these challenges, we construct a comparator LLM to update the instructions for the actor. Instead of optimizing on a sampled instance, comparator aims to identify systematic flaws throughout the structured actions/solutions.

**Step 1: Constructing positive and negative queries**. To achieve this goal, as shown in Figure 1, the comparator samples a set of data (question-answer pairs), evaluates the current action sequence on the queries, and categorizes them into well-performing (positive) and poorly-performing (negative) groups based on their performance. Specifically, we define two thresholds, $\ell$ and $h$ (where $0 < h \leq \ell < 1$), which serve as the upper and lower bounds for constructing positive and negative queries, respectively. Queries with an evaluation metric (e.g., Recall) value above $\ell$ are classified as positive, while those below $h$ are classified as negative. Based on the training dynamics, one could consider adapting the lower bound to ensure a sufficient number of negative samples for selection. After classification, we use random sampling to create a mini-batch of $b$ queries, with an equal split of positive and negative queries ($b/2$ each) for contrastive reasoning.

**Step 2: Generating instructions through contrastive reasoning**. After this, the comparator is tasked with contrasting the two groups of queries based on their key characteristics, attributing the performance gap to specific tool usage within the complex solution, and finally suggesting general modifications that can improve overall task performance. The instructions generated by the comparator are then appended to the initial prompts to update the actor.

**Insights/Justification for the comparator**. To illustrate the insights, we draw an analogy from deep neural network training, where extremely small batch sizes can introduce significant noise in gradient estimates and high variance in model updates. By adopting a batched training strategy and sampling positive and negative queries as two "mini-batches," comparator can extract a robust "gradient" to update the actor. This approach encourages comparator to generate more general and comprehensive instructions on the complex action sequence, including problem decomposition, solutions to subproblems, and the final synthesis. Moreover, as contrastive reasoning directly targets disentangling the performance gap related to input patterns and how they are handled differently by the tools, it is particularly effective in helping comparator differentiate and select tools for use. Finally, by identifying systemic flaws across a wide array of negative queries, comparator generates modifications that are not only tailored to individual samples but also to diverse data samples, enhancing generalization to novel cases.

**Demonstration example**. Figure 3 illustrates an example where comparator contrasts the patterns of positive and negative queries, identifying discrepancies in tool usage within the action sequence. It reveals that, compared to positive queries, negative queries feature more complex product descriptions, more subtle brand mentions, and additional relevant product mentions. These observations suggest: 1) an incomplete problem decomposition involving query attributes like detailed product features, 2) a potentially imprecise brand match using embedding similarity, and 3) a lack of consideration for related products in the results. Informed by these insights, actor updates its action sequence to address these subproblems and use the tools more effectively for the task, such as replacing the embedding tool with an LLM verification tool.

### 4.3 Logistic Instructions and Memory Construction

**Logistic instructions**. While instructions from the comparator are designed to improve task performance, we incorporate two types of orthogonal instructions to ensure the actions are valid and can be executed efficiently.

- **Validity check**: This instruction is triggered internally during the execution of each action. It ensures the validity of the actor's actions, such as verifying the correct use of function calls.
- **Timeout error**: To prevent inefficient action sequences that may stall the actor, we implement a timeout mechanism that triggers an error if processing exceeds a specified threshold. This error prompts the actor to adopt more efficient strategies, such as eliminating redundant operations.

**Memory Bank**. During optimization, we utilize a memory bank inspired by human decision-making processes, following Shinn et al. [41], where humans typically address current problems by analyzing the current situation and referencing past experiences. The memory bank stores tuples of action sequences, instructions from comparator, and the performance of these action sequences on a small training set (sampled from positive and negative queries). To manage the context size input to actor, we retain only the top-5 action sequences with the best performance. This memory bank enables actor to learn from both immediate instructions and historical results.

**Deployment**. At deployment, we can apply the optimized instructions or, as shown in Figure 1, the optimized actor /action sequence, which includes effective tool utilization and problem-solving strategies, to answer queries or retrieve entities. In the experiments, we demonstrate AVATAR's flexibility by applying different deployment strategies.

## 5 Experiments

**Tasks and Evaluation**. We conduct experiments on the following datasets:

- **Four challenging retrieval datasets from STARK [49] and FLICKR30K-ENTITIES [35]** to demonstrate AVATAR in handling complex real-world tasks (*cf.* details in Appendix A). For each query in the retrieval datasets, the task is to retrieve relevant entities, such as nodes in a knowledge graph or images in knowledge bases. During deployment, we directly apply the optimized action sequence to the test queries. We assess task performance by comparing the consistency of the results with the ground truth answers in the datasets, using Hit@1, Hit@5, Recall@20, and Mean Reciprocal Rank (MRR) as the metrics.
- **Three question-answering (QA) benchmarks: HotpotQA [53], ArxivQA [22], ToolQA [67]**, where the task is to provide natural language answers to the questions. We sample 100, 100, and 40 training queries, and 100, 100, and 60 testing queries for the three benchmarks, respectively. During deployment, the actor LLM uses optimized instructions to generate the action sequence for obtaining the answer. We use exact match (EM) score on HotpotQA, following previous methods. For ArxivQA and ToolQA, we use the LLM judge score for more reliable evaluation.

**Baselines**. For the knowledge retrieval tasks, we employ several embedding-based retriever models for our evaluation, following Wu et al. [49]: Dense Passage Retriever (DPR) Karpukhin et al. [17]; Vector Similarity Search methods ada-002 and multi-ada-002 using `text-embedding-ada-002` from OpenAI; and a relation-aware model, QAGNN [57], for the STARK benchmark. Additionally, we include four prevailing agent frameworks to further enrich our evaluation:

- **ReAct** [55] conducts reasoning and action in an in-context and interleaved manner to enable LLMs to interactively analyze observed information and perform actions.

Table 2: Retrieval performance (%) on STARK benchmark. Last row shows the relative improvements over the best metric value in each column.

| | AMAZON | | | | MAG | | | | PRIME | | | |
|---|---|---|---|---|---|---|---|---|---|---|---|---|
| | Hit@1 | Hit@5 | R@20 | MRR | Hit@1 | Hit@5 | R@20 | MRR | Hit@1 | Hit@5 | R@20 | MRR |
| DPR | 15.29 | 47.93 | 44.49 | 30.20 | 10.51 | 35.23 | 42.11 | 21.34 | 4.46 | 21.85 | 30.13 | 12.38 |
| QAGNN | 26.56 | 50.01 | 52.05 | 37.75 | 12.88 | 39.01 | 46.97 | 29.12 | 8.85 | 21.35 | 29.63 | 14.73 |
| ada-002 | 39.16 | 62.73 | 53.29 | 50.35 | 29.08 | 49.61 | 48.36 | 38.62 | 12.63 | 31.49 | 36.00 | 21.41 |
| multi-ada-002 | 40.07 | 64.98 | 55.12 | 51.55 | 25.92 | 50.43 | 50.80 | 36.94 | 15.10 | 33.56 | 38.05 | 23.49 |
| ReAct | 42.14 | 64.56 | 50.81 | 52.30 | 31.07 | 49.49 | 47.03 | 39.25 | 15.28 | 31.95 | 33.63 | 22.76 |
| Reflexion | 42.79 | 65.05 | 54.70 | 52.91 | 40.71 | 54.44 | 49.55 | 47.06 | 14.28 | 34.99 | 38.52 | 24.82 |
| AVATAR-C | 40.92 | 63.63 | 53.68 | 51.73 | 33.25 | 52.17 | 47.88 | 41.34 | 8.82 | 23.82 | 30.32 | 16.20 |
| AVATAR | **49.87** | **69.16** | **60.57** | **58.70** | **44.36** | **59.66** | 50.63 | **51.15** | **18.44** | **36.73** | **39.31** | **26.73** |
| Relative Improvement | 16.6% | 6.3% | 9.9% | 12.2% | 9.6% | 2.1% | -0.3% | 8.7% | 20.7% | 5.0% | 2.1% | 7.7% |

- **Reflexion** [41] uses self-reflection on the current task completion and stores these reflections in an episodic memory buffer to enhance decision-making in subsequent trials.

- **ExpeL** [61] extracts insights from successful and failed action sequences, retrieving and including them in the context during inference. We apply ExpeL on the QA datasets and, due to its high cost on large-scale retrieval tasks, compare it with AVATAR on a sampled STARK-MAG test set.

- **Retroformer** [56] reinforces LLM agents and automatically tunes their prompts by learning a retrospective model through policy gradient. We compare the performance of AVATAR with the reported result by Retroformer on HotpotQA due to the additional training involved.

We include an ablation model, AVATAR-C, which removes the comparator from our optimization pipeline. This comparison aims to validate the effectiveness of the comparator. The LLM version information is provided in Appendix B.

**Function library**. For the knowledge retrieval tasks, our function library consists of twenty-eight functions that facilitate access to, operation on, and reasoning over the knowledge information by LLM agents. For the QA tasks, we provide web search tools such as Google and Arxiv search APIs. See Appendix E for details. We used the same function library across all agent methods.

**General pipeline**. For AVATAR, we optimize the agent for a fixed number of epochs and select the action sequence or instruction with the highest performance. We use the same initial prompt structure, the metric Recall@20 or Accuracy for constructing positive and negative queries, and hyperparameters ($\ell = h = 0.5, b = 20$) for all datasets.

## 5.1 Textual and Relational Retrieval Tasks

We employ the AMAZON, MAG, and PRIME datasets from the STARK benchmark [49], a large-scale semi-structured retrieval benchmark that integrates textual and relational knowledge (*cf.* detailed description in Appendix A). Here, the entities to be retrieved are defined as nodes in a graph structure, with knowledge associated with each entity including both textual descriptions and relational data. We use the official splits from the STARK benchmark.

**Takeaway 1: AVATAR outperforms state-of-the-art models**. Table 3 shows that AVATAR substantially outperforms leading models such as Reflexion across all metrics on the STARK benchmark. Notably, the average improvement of AVATAR is 15.6% on Hit@1 and 9.5% on MRR. ReAct agents, however, cannot optimize based on instructions for improved tool usage and tend to select tools based on the LLM's prior knowledge, which may not be optimal for the given task. We observe that ReAct agents apply similar tools across various queries and struggle to explore alternative tool usage even with extensive in-context reasoning. Results for agent methods using GPT-4 Turbo are provided in Appendix B, showing similar conclusions. For comparison with ExpeL, the results in Table 6 show that it performs similarly to ReAct, underperforming AVATARby a large margin.

**Takeaway 2: Comparator greatly impacts the actor's performance**. The comparison of AVATAR with its ablation variant, AVATAR-C, highlights the significant advantages of the comparator module. Although AVATAR-C conducts validity and timeout checks, integrating Comparator into AVATAR adds a comprehensive instruction mechanism crucial for identifying clear directions to improve the agents, underlining comparator's key role in optimizing actor.

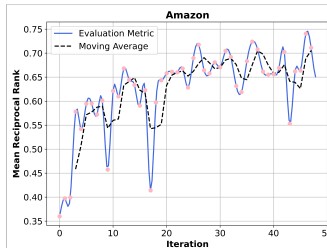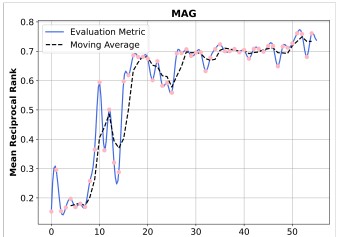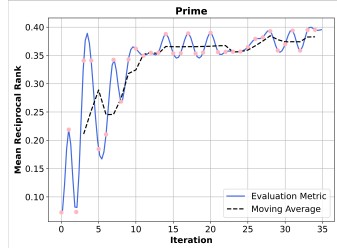

Figure 4: **Optimization dynamics of AVATAR agents on STARK**. The figures show validation performance (solid line) and its moving average (dashed line) during the optimization of AVATAR.

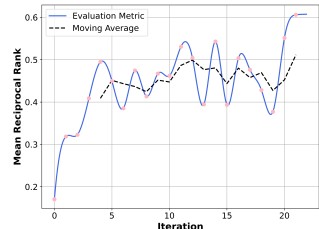

| | Hit@1 | Hit@5 | R@20 | MRR |
|---|---|---|---|---|
| clip-vit-large-patch14 | 37.2 | 56.4 | 72.8 | 46.3 |
| ReAct (claude3) | 38.8 | 54.8 | 71.6 | 46.1 |
| Reflexion (claude3) | 28.4 | 53.2 | 75.2 | 41.2 |
| AVATAR-C (claude3) | 28.8 | 53.2 | 78.4 | 40.0 |
| AVATAR (claude3) | **42.4** | **63.0** | **79.2** | **52.3** |
| Relative Improvement | 9.2% | 11.7% | 5.3% | 13.0% |

Figure 5: Performance (left) and AVATAR's optimization dynamics (right) on FLICKR30K-ENTITIES.

**Takeaway 3: AVATAR effectively improves agents during optimization**. Figure 4 illustrates the agents' performance on the validation set during optimization. Impressively, AVATAR agents show significant performance improvements, e.g., from 35% to 75% on AMAZON and from 20% to 78% on MAG. This evidence strongly supports the effectiveness of the instructions generated by our comparator. Additionally, our memory bank, which stores past best-performing actions, encourages AVATAR agents to gradually converge by the end of the optimization process.

**Takeaway 4: AVATAR can generalize to real-world tasks**. Comparator generates instructions tailored to groups of retrieval queries, promoting generalizable modifications for novel queries. We validate this capability by applying optimized actions to human-generated leave-out queries from the STARK benchmark, which differ notably from the training data used to optimize our agents. Results in Table 5 (Appendix B) show that AVATAR significantly outperforms other models, achieving an average improvement of 20.9% on Hit@1. Further, in another study of Appendix B, we assess AVATAR's robustness to hyperparameters $h$ and $\ell$, showing that it maintains stable performance and generalization across different parameter values.

## 5.2 Image Retrieval Task

We further experiment on FLICKR30K ENTITIES [35], an image retrieval dataset of 30k images with annotated bounding boxes and descriptive phrases (Appendix A). In Table 2, AVATAR again shows significant improvements. In contrast, Reflexion agents struggle with "overfitting," where they are easily misled by specific image data, leading to inappropriate actions (e.g., trying to "extract the color of a hat" from images without hats). AVATAR effectively avoids such pitfalls through batch-wise contrastive reasoning, which provides a broader perspective.

**Takeaway 5: AVATAR generates impressive and generalizable actions**. In Figure 5 (left), the final actions of the AVATAR agent detailed in Figure 8 (Appendix B), achieve advanced performance. Notably, AVATAR skillfully manages input queries and leverages Inverse Document Frequency (IDF) scores to refine phrase matching, ultimately synthesizing accurate answers. Beyond using existing tools, AVATAR agents can develop high-level tools, such as IDF-based reweighting, suggesting a promising direction for dynamic tool libraries and enhanced tool generation.

**Takeaway 6: Emerging behaviors during optimization**. In Figure 6, we present concrete cases illustrating key interactions between actor and comparator. In each instance, comparator identifies critical flaws, including information omission, ineffective tool usage, and suboptimal synthesis of varying scores. The instructions subsequently prompt actor to enhance retrieval strategies, tool selection, and precise score combinations. Furthermore, frequent references to tool usage underscore comparator's focused examination of tool utilization during optimization.

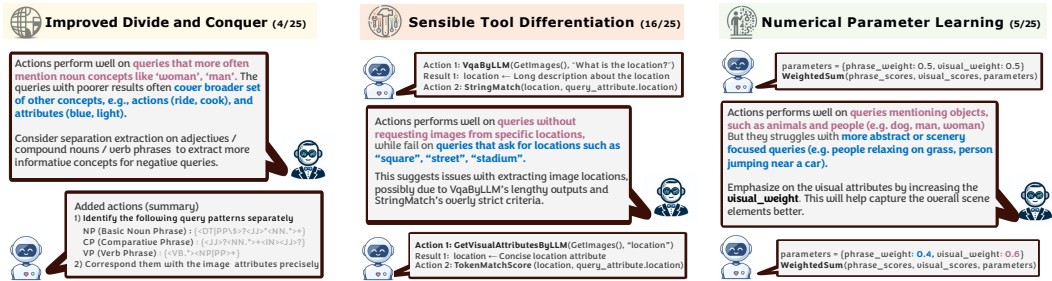

Figure 6: **Representative instruction types from the comparator.** We provide three cases where the comparator guides the actor towards (1) better divide-and-conquer strategies for multi-step problem-solving, (2) more sensible differentiation between good and bad tool usage/combinations, and (3) adjustments in the weights to generate the final answers. We record the number of occurrences $X$ under each instruction type over 25 iterations on FLICKR30K-ENTITIES, indicated by ($X$/25).

Table 3: Performance (%) on three QA benchmarks. Last row shows the relative improvements over the best metric value in each column.

|  | HOTPOTQA | ARXIVQA | TOOLQA | | | |
|---|---|---|---|---|---|---|
|  |  |  | SCIREX-EASY | SCIREX-HARD | AGENDA-EASY | AGENDA-HARD |
| CoT | 28.0% | 58.0% | 1.7% | 0.0% | 0.0% | 0.0% |
| ReAct | 40.0% | 72.0% | 31.7% | 17.5% | 38.3% | 3.33% |
| Reflexion | 46.0% | 77.0% | 28.3% | 13.3% | 30.0% | 3.33% |
| ExpeL | 39.0% | 73.0% | 36.7% | 14.5% | 56.6% | 1.67% |
| Retroformer (#retry=1) | 51.0% | - | - | - | - | - |
| AVATAR-C | 41.0% | 73.0% | 31.7% | 13.3% | 31.7% | 1.67% |
| AVATAR | **53.0%** | **84.0%** | **37.5%** | **23.3%** | **60.0%** | **4.17%** |
| Relative Improvement | 3.92% | 9.09% | 2.18% | 33.1% | 5.82% | 25.0% |

## 5.3 Question Answering Tasks

Finally, we applied AVATAR to three widely used QA benchmarks. For ToolQA, we tested AVATAR and the baselines on two different domains: SciREX, which focuses on extracting information from full-length machine learning papers, and Agenda, which involves personal agenda-related questions. Both datasets have easy and hard versions.

**Takeaway 7: AVATAR outperforms on QA tasks by offering better context understanding**.
Table 3 shows that AVATAR consistently outperforms state-of-the-art methods across all three QA datasets, with especially strong results on TOOLQA. In SCIREX-HARD, which focuses on extracting complex information from long scientific papers, AVATAR shows a 33.1% improvement, while in AGENDA-HARD, it achieves a 25.0% relative gain. These improvements are attributed to AVATAR's ability to generate optimized prompts that help the agent better understand the broader patterns and contexts of the questions, leading to more accurate answers and improved generalization across question types, from simple to complex.

## 6 Conclusion and Future Work

In this study, we introduce AVATAR, a novel framework that automates the optimization of LLM agents for enhanced tool utilization in multi-step problems, focusing on complex retrieval and QA tasks. AVATAR demonstrates remarkable improvements across seven diverse datasets. This success can largely be attributed to the comparator module, which effectively refines agent performance through the iterative generation of holistic and strategic prompts. A key innovation of comparator is its use of contrastive reasoning with batch-wise sampling, enabling it to identify systemic flaws and extract robust "gradients" for comprehensive agent improvement across diverse scenarios. While we observe substantial progress from AVATAR, we discuss its limitations in Appendix D regarding its scalability *etc.*Future work can explore extending this methodology to other challenging agent tasks, visual reasoning tasks, and more dynamic environments, or designing better memory banks for dynamically storing knowledge and experience from past training.

## Acknowledgement

We thank lab members in Zou and Leskovec's labs for discussions and for providing feedback on our manuscript. We also gratefully acknowledge the support of DARPA under Nos. N660011924033 (MCS); NSF under Nos. OAC-1835598 (CINES), CCF-1918940 (Expeditions), DMS-2327709 (IHBEM); Stanford Data Applications Initiative, Wu Tsai Neurosciences Institute, Stanford Institute for Human-Centered AI, Chan Zuckerberg Initiative, Amazon, Genentech, GSK, Hitachi, SAP, and UCB. The content is solely the responsibility of the authors and does not necessarily represent the official views of the funding entities.

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

# A  Retrieval Tasks

**STARK**. On the STARK benchmark, we are given a relation-text knowledge base, based on a knowledge graph $G = (V, E)$ and a collection of free-text documents $D$. We represent the relation-text knowledge base of size $n$ as $\mathcal{E} = \{(v_i, d_i, g_i)\}_{i=1}^n$, where $v_i \in V$ represents a node on the knowledge graph, $d_i \in D$ is the text document related to the node, and $g_i$ is the connected component of $G$ containing $v_i$.

The query set $Q$ in STARK is derived from both $G$ and $D$, where each $q_i \in Q$ contains requirements based on $d_i$ and $g_i$. The answer set $A_i$, which includes $v_i$, is a set of nodes satisfying both relational and textual requirements. The task on STARK is defined as follows: Given the knowledge base $\mathcal{E}$ consisting of relational and textual information, and a text query $q_i$, the output is a set of nodes $A_i$ such that $\forall a_i \in A_i$, $a_i$ satisfies the relational requirements in the knowledge graph and textual requirements in its text documents.

**FLICKR30K ENTITIES**. On the FLICKR30K ENTITIES dataset, we are given an image-text knowledge base. We denote an image-text knowledge base of size $n$ as $\mathcal{E} = \{(v_i, q_i, T_i)\}_{i=1}^n$. Sample $i$ consists of an image $v_i$, its descriptive caption $q_i$, and entity bounding box information $T_i$. Specifically, $T_i = \{(c_{ij}, p_{ij})\}_{j=1}^{b_i}$, where $b_i$ represents the number of bounding boxes annotated in image $i$, $c_{ij}$ is the coordinate of the $j$-th bounding box, and $p_{ij}$ describes the entity in the corresponding bounding box.

In our task, the image captions serve as the text query; therefore, all $q_i$ in the dataset are not accessible to the agent to prevent information leakage. However, the agent can access $v_i$ and $T_i$ to fully utilize the vision and language information. The task on FLICKR30K ENTITIES is defined as follows: Given the knowledge base $\mathcal{E}$ with images and bounding box information, and a text query $q_i$, the output is an image $v_i$ that satisfies the visual requirements in the image and textual requirements in the corresponding bounding boxes $T_i$.

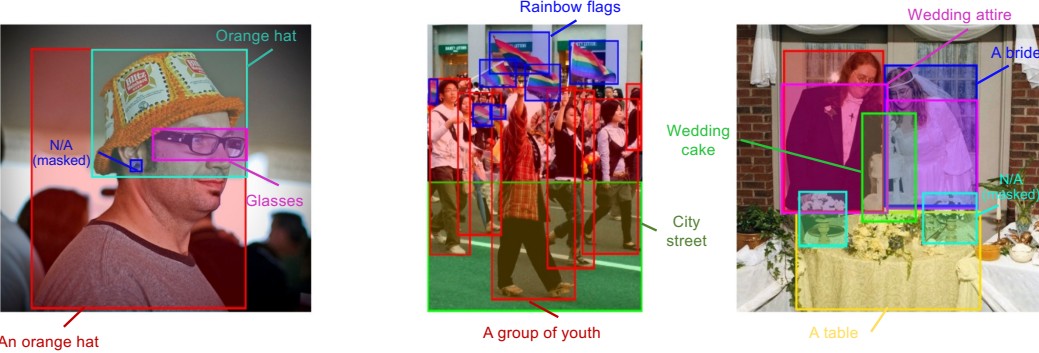

Figure 7: **Example data on FLICKR30K ENTITIES.** Each entity is an image along with its image patches and associated phrases with the image patches.

# B  Experiment Details and Additional Results

## B.1  Experiment Setup

**LLM versions for agent methods**.

- For the knowledge retrieval tasks, we use `claude-3-opus` as the backbone LLM in the main paper by default, and report results using `gpt-4-turbo` in Appendix B due to space limitations.

- For the QA tasks, we use `gpt-4` for HotpotQA for fair comparison with previous methods and `gpt-4o` for the other two QA datasets.

Table 4: Retrieval performance (%) on STARK benchmark. Last row shows the relative improvements over the best metric value among the baselines.

| | AMAZON | | | | MAG | | | | PRIME | | | |
|---|---|---|---|---|---|---|---|---|---|---|---|---|
| | Hit@1 | Hit@5 | R@20 | MRR | Hit@1 | Hit@5 | R@20 | MRR | Hit@1 | Hit@5 | R@20 | MRR |
| DPR (roberta) | 15.29 | 47.93 | 44.49 | 30.20 | 10.51 | 35.23 | 42.11 | 21.34 | 4.46 | 21.85 | 30.13 | 12.38 |
| QAGNN (roberta) | 26.56 | 50.01 | 52.05 | 37.75 | 12.88 | 39.01 | 46.97 | 29.12 | 8.85 | 21.35 | 29.63 | 14.73 |
| ada-002 | 39.16 | 62.73 | 53.29 | 50.35 | 29.08 | 49.61 | 48.36 | 38.62 | 12.63 | 31.49 | 36.00 | 21.41 |
| multi-ada-002 | 40.07 | 64.98 | 55.12 | 51.55 | 25.92 | 50.43 | **50.80** | 36.94 | 15.10 | 33.56 | 38.05 | 23.49 |
| ReAct (gpt4) | 38.83 | 62.50 | 50.39 | 49.16 | 23.50 | 46.50 | 43.11 | 33.91 | 10.83 | 30.83 | 32.16 | 19.39 |
| Reflexion (gpt4) | 41.45 | 64.83 | 53.98 | 52.22 | 33.44 | 51.33 | 49.14 | 41.34 | 14.27 | 35.11 | 39.29 | 23.61 |
| Reranker (gpt4) | 44.79 | 71.17 | 55.35 | 55.69 | 40.90 | 58.18 | 48.60 | 49.00 | 18.28 | 37.28 | 34.05 | 26.55 |
| AVATAR-C (gpt4) | 32.03 | 58.46 | 54.03 | 44.00 | 25.97 | 45.62 | 46.68 | 35.12 | 9.52 | 26.04 | 32.62 | 17.58 |
| **AVATAR (gpt4)** | **48.82** | **72.03** | 56.04 | 57.17 | **46.08** | 59.32 | 49.70 | **52.01** | 20.10 | 39.89 | 42.23 | 29.18 |
| Relative Improvement (over Best Baseline) | 9.0% | 1.2% | 1.3% | 2.7% | 12.7% | 2.1% | -2.2% | 6.1% | 10.0% | 7.0% | 11.0% | 9.9% |

## B.2 Additional Experimental Results

**(1) AVATAR results on STARK using GPT-4 Turbo (0125) as LLM backbone**. In Table 4, we provide the results on STARK using GPT-4 Turbo (0125) as the backbone LLM.

Table 5: Retrieval performance (%) on the leave-out sets of human-generated queries in STARK.

| | AMAZON | | | | MAG | | | | PRIME | | | |
|---|---|---|---|---|---|---|---|---|---|---|---|---|
| | Hit@1 | Hit@5 | R@20 | MRR | Hit@1 | Hit@5 | R@20 | MRR | Hit@1 | Hit@5 | R@20 | MRR |
| DPR (roberta) | 16.05 | 39.51 | 15.23 | 27.21 | 4.72 | 9.52 | 25.00 | 7.90 | 2.04 | 9.18 | 10.69 | 7.05 |
| ada-002 | 39.50 | 64.19 | 35.46 | 52.65 | 28.57 | 41.67 | 35.95 | 35.81 | 17.35 | 34.69 | 41.09 | 26.35 |
| multi-ada-002 | 46.91 | 72.84 | 40.22 | 58.74 | 23.81 | 41.67 | **39.85** | 31.43 | 24.49 | 39.80 | 47.21 | 32.98 |
| ReAct | 45.65 | 71.73 | 35.95 | 58.81 | 27.27 | 40.00 | 35.95 | 33.94 | 21.73 | 33.33 | 41.09 | 28.20 |
| Reflexion | 49.38 | 64.19 | 35.95 | 58.96 | 28.57 | 39.29 | 35.95 | 36.53 | 16.52 | 33.03 | 41.09 | 23.99 |
| AVATAR | **58.32** | **76.54** | **42.43** | **65.91** | **33.33** | **42.86** | 35.94 | **38.62** | **33.03** | **51.37** | **53.34** | **41.00** |
| Rel. Impr. | 17.5% | 5.1% | | 11.8% | 16.7% | 2.9% | | 5.7% | 28.7% | 27.3% | | 21.4% |

**(2) AVATAR results on STARK's human-generated splits**. In Table 5, we demonstrate AVATAR's ability to generalize to test queries with distributions different from the question-answering pairs used to optimize the actor agents.

Table 6: Performance metrics for different models on the subset of the STARK-MAG dataset.

| | MAG (#Test=50) | | | |
|---|---|---|---|---|
| | Hit@1 | Hit@5 | Recall@20 | MRR |
| DPR | 16.00 | 40.00 | 51.84 | 27.39 |
| QAGNN | 20.00 | 52.00 | 49.71 | 36.39 |
| ada-002 | 40.00 | 58.00 | 55.93 | 47.76 |
| multi-ada-002 | 32.00 | 58.00 | 58.81 | 43.58 |
| ReAct | 46.00 | 60.00 | 54.67 | 50.92 |
| ExpeL | 40.00 | 58.00 | 55.94 | 47.43 |
| Reflexion | 48.00 | 64.00 | 57.43 | 52.31 |
| AvaTaR-C | 44.00 | 60.00 | 52.49 | 50.16 |
| AvaTaR | 52.00 | 64.00 | 53.86 | 56.74 |
| Relative Improvement | 8.33% | 0.00% | -8.42% | 8.48% |

**(3) AVATAR results and comparison with ExpeL on STARK-MAG subset**. In Table 6, AVATAR demonstrates consistently higher performance than ExpeL across most metrics, notably achieving the highest Hit@1 and MRR scores. While ExpeL performs well in Recall@20, AVATAR 's overall improvements highlight its superior capability in precise retrieval tasks and tool-assisted knowledge retrieval.

**(4) Final action sequence by AVATAR on FLICKR30K-ENTITIES**. In Figure 8, we present the final actions optimized by AVATAR on FLICKR30K-ENTITIES.

**(5) Sensitivity of AVATAR to upper and lower bounds**. We evaluated various combinations of $\ell$ and $h$, focusing on the STARK-AMAZON dataset due to computational constraints. Table 7 presents the Hit@1 results for different $\ell$ and $h$ values.

```
Input: Any query (example: "A man with pierced ears is wearing glasses and an orange hat .");
        Action Space: {GetImages, GetEmbeddingSimilarity, GetVisualAttributesByLLM, , ...]
Output: Retrieved Image IDs

✅ Remove empty spaces or non-alphabetic characters
Action 1: CleanQueryText[query]
Result 1: normalized_query ← "a man with pierced ears is wearing glasses and an orange hat"

✅ Get all phrases from the knowledge base
Action 2: GetBagofPhrases()
Result 2: phrases_list ← [["a man", "grass", "sky"], ["a", "cat", ...]]

✅ Compute IDF for phrase importance
Action 3: ComputeIDFScores[Flatten[phrases_list]]
Result 3: idf_scores ← {"pierced": 0.5, "man": 0.0012, ...}

✅ Get visual attributes for the candidate images
Action 4: GetVisualAttributesByLLM[GetImages(), ["color", "object", "action", "count"]]
Result 4: visual_attributes ← {node_id_1: {"color": "red", ...}, node_id_2: {...}, ...}

✅ Evaluate textual and visual relevance
Action 5: [','.join(list) for list in phrases_list]
Result 5: phrase_sentences ← ["a man, grass, sky", "a, cat, playground",...]

✅ Evaluate textual and visual relevance
Action 6: ComputeEmbeddingSimilarity[normalized_query, phrase_sentences]
Result 6: text_scores

Action 7: ComputeEmbeddingSimilarity[normalized_query, visual_attributes]
Result 7: visual_scores

✅ Match query phrases with node attributes using IDF scores
Action 8: MatchQueryPhrases[normalized_query.split(), visual_attributes]
Result 8: phrase_match_scores

✅ Reweight the phrase_match_scores with IDF score
Action 9: ReweightByIDFScore[phrase_match_scores, idf_scores]
Result 9: reweighted_match_scores

✅ Aggregate scores with weighted parameters
Action 10: WeightedSum[text_scores, visual_scores, reweighted_match_scores, weights=(0.5, 0.3, 0.2)]
Result 10: aggregated_scores

✅ Normalize scores for final ranking
Action 11: NormalizeScores[aggregated_scores]
Result 11: normalized_scores = {node_id: normalized_score, ...}

Final Result: answers = GetTopkEntities[normalized_scores, k=5]
✅ Excellent task performance
```

Figure 8: **Optimized Action Sequence by AVATAR on FLICKR30K-ENTITIES.**.

Table 7: Hit@1 results for different combinations of $\ell$ and $h$ values on the STARK-AMAZON dataset.

|              | $h = 0.3$ | $h = 0.4$ | $h = 0.5$ |
|--------------|-----------|-----------|-----------|
| $\ell = 0.5$ | 48.32     | 50.01     | 49.87     |
| $\ell = 0.6$ | 47.89     | 49.56     | **50.45** |
| $\ell = 0.7$ | 47.75     | 48.56     | 49.34     |

**Key Observations**

- The framework exhibits **robustness** to variations in $\ell$ and $h$, with Hit@1 fluctuations limited to a range of 2.7%.

- A **performance decline** is observed when the gap between $\ell$ and $h$ becomes too large, potentially due to the exclusion of certain training queries that fall within the $(h, \ell)$ interval.

- A **moderate gap** between $\ell$ and $h$ leads to slight performance improvements, suggesting that a balanced separation between positive and negative queries can enhance pattern differentiation without compromising the number of training queries.

The results indicate that setting $\ell = 0.6$ and $h = 0.5$ yields an improved Hit@1 score compared to the baseline reported in the original paper. Overall, this analysis underscores the robustness of the framework, which relies on a minimal set of hyperparameters, including $\ell$, $h$, batch size $b$, and training epochs.

## C  Prompts

We keep only two prompt templates for our framework on all tasks: (1) The prompt template given to actor as initially instructions, and (2) the prompt template given to the comparator to conduct contrastive reasoning and generate the instructions for the actor. Below are the complete templates:

This is the prompt given to actor as initially instructions:

```
You are an expert user of a knowledge base, and your task is to answer a set of
    ↪ queries. I will provide your with the schema of this knowledge base:
<knowledge_base_schema>

You have access to several APIs that are pre-implemented for interaction with the
    ↪ knowledge base:
<func_call_description>

Information of queries: Below are several query examples that you need to carefully
    ↪ read through:
"
<example_queries>
"

Task: Given an input query, you should write the actions in Python code to calculate
    ↪  a 'node_score_dict' for <n_init_candidates> node IDs, which are input as a
    ↪ list. These node IDs, referred to as 'candidate_ids', are a subset of node
    ↪ IDs from the knowledge base, and the nodes belong to the type(s) <
    ↪ candidate_types>. In 'node_score_dict: Dict[int, float]', each key should be
    ↪  a node ID, and each value should be the corresponding node score. This
    ↪ score should indicate the likelihood of the node being the correct answer to
    ↪  the query.

Output format: Firstly, you should establish a connection between the given queries
    ↪ and the query patterns to the schema of the knowledge base. Secondly,
    ↪ generate an outline for the code that will compute the scores for all the
    ↪ candidate nodes provided in the query examples. Finally, develop the main
    ↪ function named 'get_node_score_dict', which takes two required parameters: '
    ↪ query' and 'candidate_ids', and optional parameters declared in '
    ↪ parameter_dict'. Note that 'parameter_dict' is a dictionary of parameters
    ↪ and their default values where you can declare any parameters or weights
    ↪ used during computing the node scores. If no optional parameters are needed,
    ↪  leave 'parameter_dict' as an empty dictionary. Overall, your output should
    ↪ follow the structure:

'''python
# <code outlines>
import <package1>
...

parameter_dict = {<parameter_name1>: <default_value1>,
                  <parameter_name2>: <default_value2>,
                  ...}

def get_node_score_dict(query, candidate_ids, **parameter_dict):
    node_score_dict = {}
    # your code
    return node_score_dict
'''

Hints:
- Observe the example queries carefully and consider the key attributes to extract.
- Use '''python and ''' to wrap the complete code, and do not use any other
    ↪ delimiters.
- You can use any of the pre-implemented APIs but should avoid modifying them.
- You can include other functions besides 'get_node_score_dict', but ensure they are
    ↪  fully implemented.
```

```
- The code should be complete without placeholders and dummy functions.
- Optimize the integrity of the code, e.g., corner cases.
- Minimize computational expenses by early elimination of candidate nodes that don't
    ↪   meet relational requirement (if any).
- Avoid conducting unnecessary and redundant computations, especially when using
    ↪   loops.
- Make use of 'parameter_dict' to avoid hard-coding parameters and weights.
- Use the functions that end with 'by_llm' wisely for more accurate searches.
- Use 'debug_print' smartly to print out any informative intermediate results for
    ↪   debugging.
- Exclude or comment out any example uses of 'get_node_score_dict' in the output
    ↪   code.

Your output:
```

This is the prompt given to comparator to generate the instructions for the actor:

```
<initial_prompt>

<previous_actions>

After executing the above actions on user queries, some queries have yielded good
    ↪   results, while others have not. Below are the queries along with their
    ↪   corresponding evaluation metrics:
Well-performing queries:
<positive_queries_and_metric>
Poorly-performing queries:
<negative_queries_and_metric>

Task:
(1) Firstly, identify and contrast the patterns of queries that have achieved good
    ↪   results with those that have not.
(2) Then, review the computational logic for any inconsistencies in the previous
    ↪   actions.
(3) Lastly, specify the modification that can lead to improved performance on the
    ↪   negative queries. You should focus on capturing the high-level pattern of
    ↪   the queries relevant to the knowledge base schema.
```

# D   Limitations

We identify several potential limitations of our work:

- **Scalability**: AvaTaR is designed to scale with large language models (LLMs) that support extended context lengths (up to 128k tokens), enabling it to handle numerous tools and complex tasks. However, increased latency and other practical limitations may hinder performance in scenarios requiring hundreds of tools or high complexity. Future research could focus on incorporating specialized, tool-augmented LLMs as auxiliary agents to facilitate smoother scaling.

- **Computation Requirements**: Managing longer contexts and multiple tool interactions within AvaTaR increases computational demands, which can significantly raise operational costs. These requirements necessitate substantial resources to maintain efficient performance, particularly when scaling to larger datasets or more intricate tasks.

- **Potential Failure Modes**: Although AvaTaR performs well on known queries, its performance may diminish when faced with queries that require new or unfamiliar combinations of tools. This limitation could be mitigated by integrating adaptive learning techniques and continuous monitoring, which would allow AvaTaR to better handle novel tool requirements.

# E Function library

## E.1 Complex Retrieval Tasks

Please refer to Table 8 and Table 9 for the detailed functions.

## E.2 General QA Tasks

For general QA tasks, we use the following tools:

- `WEB_SEARCH`: A general-purpose tool that performs web searches to answer questions. Useful for retrieving up-to-date information from the internet when other sources are unavailable.
- `ARXIV_SEARCH`: This tool retrieves information about academic papers from Arxiv using a paper's unique ID. This function call can provide metadata and other details for academic references.
- `Wiki_SEARCH`: If you have a question or name to lookup, this tool uses a Wikipedia search to retrieve relevant information.
- `RETRIEVE_FROM_DB`: This tool is used to retrieve relevant information from a database. This is only available on ToolQA.

| Function Name | Input | Output |
|---|---|---|
| ParseAttributeFromQuery | query: The string to be parsed, attributes: The list of attributes to be extracted from the query | This function parses a 'query' into a dictionary based on the input list 'attributes' |
| GetTextEmbedding | string: The array of list to be embedded | Embeds N strings in a list into N tensors |
| GetRelevantChunk | query: The input query string, node_id: The ID of the node | Get the relevant chunk of information for the node based on the query |
| GetFullInfo | node_id: The ID of the node | Get the full information of the node with the specified ID |
| GetEntityDocuments | node_id: The ID of the node | Get the text information of the node with the specified ID |
| GetRelationInfo | node_id: The ID of the node | Get the relation information of the node with the specified ID |
| GetRelationDict | node_id: The ID of the node | Get the relation dictionary for the node with the specified ID, where the keys are relation type and values are neighbor nodes. |
| GetRelatedEntities | node_id: The ID of the node | Get the nodes related to the specified node |
| GetEntityIdsByType | type: The type of node to retrieve | Get the IDs of nodes with the specified type |
| GetEntityTypes | node_id: The ID of the node | Get the type of the node with the specified ID |
| GetEntityEmbedding | node_ids: An array of candidate node ids to be embedded | Get the embedding indices of nodes with ID 'node_ids' |
| ComputingEmbeddingSimilarity | embedding_1 and embedding_2 | The cosine similarity score of two embeddings |
| ComputeQueryEntitySimilarity | query: The input query string, node_ids: An array of candidate node id to be compared with the query | Compute embedding similarity between 'query' (str) and the nodes' in 'node_ids' (list) |
| ComputeExactMatchScore | string: The string to be matched, node_ids: The list of candidate node id to be compared with the string | For each node in 'node_ids', compute the exact match score based on whether 'string' is included in the information of the node |
| TokenMatchScore | string: The string to be matched, node_ids: The list of candidate node id to be compared with the string | For each node in 'node_ids', computes recall scores between 'string' and the full information of the node |
| SummarizeTextsByLLM | texts: The list of texts to be summarized | Use LLM to summarize the provided texts |
| ClassifyEntitiesByLLM | node_ids: The array of candidate node ids to be classified, classes: The list of classes to be classified into | Use LLM to classify each node specified by 'node_ids' into one of the given 'classes' or 'NA' |
| ClassifyByLLM | texts: The list of texts to be classified, classes: The list of classes to be classified into | Use LLM to classify each text into one of the given 'classes' or 'NA' |
| ExtractRelevantInfoByLLM | texts: The list of texts to extract info from, extract_term: the terms to identify relevant information | Use LLM to extract relevant information from the texts based on extract_term, return sentences or 'NA' |
| CheckRequirementsByLLM | node_ids: The array of candidate node ids to be checked, requirement: The requirement to be checked | Use LLM to check if node(s) with 'node_ids' satisfies to 'requirement' |
| GetSatisfictionScoreByLLM | node_ids: The array of candidate node ids to be scored, query: The input query from user | Use LLM to score the node with 'node_ids' based on the given 'query' |
| FINISH | final_reranked_answer_list: The final answer | This function is used to indicate the end of the task |

Table 8: Function library on STARK

| Function Name | Input | Output |
|---|---|---|
| ParseAttributeFromQuery | query: The string to be parsed, attributes: The list of attributes to be extracted from the query | This function parses a 'query' into a dictionary based on the input list 'attributes' |
| GetBagOfPhrases | image_ids: The image id array to get the phrases from | Returns a list of phrase list for each image in the image_ids list |
| GetEntityDocuments | image_ids: The image id array to get the text information from | Returns a list of text information for each image in the image_ids list |
| GetClipTextEmbedding | string: The list of strings to be embedded | Embed a string or list of N strings into N embeddings |
| GetPatchIdToPhraseDict | image_ids: The image list to get the patch_id to phrase list dictionary from | Returns a list of patch_id to phrase list dictionary for each image |
| GetImages | image_id_lst: The list of image ids | Return a list of images with corresponding ids |
| GetClipImageEmbedding | image_lst: The list of images to be embedded | Embed the images of a list of N image_ids into N tensors |
| GetImagePatchByPhraseId | image_id: the id of an image, patch_id: the patch id on the image | Return the patch image for the given image_id and patch_id |
| ComputingEmbeddingSimilarity | embedding_1 and embedding_2 | The cosine similarity score of two embeddings |
| ComputeF1 | string_to_match: The key word to be matched, strings: The list of strings to be calculated f1 score with the key word | Compute the F1 score based on the similarity between 'string_to_match' and each string in 'strings' |
| TokenMatchScore | string_to_match: The key word to be matched, strings: The list of strings to be calculated recall score with the key word | Compute the recall score based on the similarity between 'string_to_match' and each string in 'strings' |
| ComputeExactMatchScore | string_to_match: The key word to be matched, strings: The list of strings to be exact matched with the key word | Compute the exact match score based on whether 'string_to_match' is exactly the same as each string in 'strings' |
| VqaByLLM | question: The question to be answered, image_lst: The list of images | Use LLM to answer the given 'question' based on the image(s) |
| ExtractVisualAttributesByLLM | attribute_lst: The list of attributes to be extracted, image_lst: The list of images | Use LLM to extract attributes about the given 'attribute_lst' from each image |
| FINISH | final_reranked_answer_list: The final answer | This function is used to indicate the end of the task |

<p align="center">Table 9: Function library on Flickr30K Entities</p>

