# OpenReview forum: "AvaTaR: Optimizing LLM Agents for Tool Usage via Contrastive Reasoning"
_NeurIPS.cc/2024/Conference — NeurIPS 2024 poster_

### Official Review · Reviewer_JqLP · 2024-07-09

**Soundness:** 2
**Presentation:** 3
**Contribution:** 2
**Rating:** 4
**Confidence:** 5

**Summary:**

This paper introduces a new framework, AVATAR, that allows LLM agents to optimize the performance of Knowledge Retrieval. In the framework, a Comparator agent is adopted to extract insight from the positive and negative samples. The experiments on four retrieval datasets show the effectiveness of the method.

**Strengths:**

1. The schematics in the paper are well drawn and demonstrate the proposed method well.
2. The experimental settings and prompts are detailed.
3. The experiments are performed on STARK and Flickr30K Entities benchmarks to show the effectiveness of the methods.

**Weaknesses:**

The main weakness is the lack of innovation and comparison compared with cutting-edge works.

1, This proposed AVATAR is very similar to ExpeL [1] in terms of method and has no obvious innovation in comparison. ExpeL adopts a Reflexion agent (actor) to gather success and failure experiences of multi-step tasks, which is the same as the actor's role in this paper. ExpeL adopts another agent to compare a failed trajectory with a successful trajectory for the same task and extract insights from the comparison, which is the same as the instruction generation process of the comparator in this paper. ExpeL adopts the experience pool to retrieve successful trajectories, which is the same as the Memory Bank in this paper. Similarly, Autoguide [2] also extracts insights from experiences. Therefore, the method of auto-optimizing the instruction by comparing the successful and failed trajectories has no innovation compared with Expel [1] and Autoguide [2]. The author should conduct more surveys and read relevant cutting-edge works.

2, The paper claims their method achieves SOTA on STARK and FLICKR30K-ENTITIES benchmarks, but they only compared with relatively basic methods. The author employs several outdated embedding-based retrievers as the comparison. However, text-to-image retrieval on FLICKR30K-ENTITIES has long been dominated by the vision-language model [3] [4]. Internvl [3] and Beit [4] achieve high recall@1 (from 80 to 90) on this benchmark. Therefore, it is unreasonable to claim that the proposed method achieves the retrieval SOTA on these two benchmarks.

3, The results of the comparison on FLICKR30K-ENTITIES are missing. Quantitative results on FLICKR30K-ENTITIES are only shown in Figure 5 (right), but the results of all methods on this benchmark are not found in the paper. Considering that STaRK is a relatively new benchmark, not many methods have experimented on it, so results and analysis on recognized benchmarks are important.

4, Many sentences in the paper are too obscure to understand and lack clear explanations. For example, "these per-sample instructions tend to be narrow and fail to identify flaws across all components of a complex solution. Additionally, while certain tool combinations may be effective for one type of input, their effectiveness can vary with others, leading to decreased performance when applied across different scenarios." in lines 152-154, what do "per-sample instructions" mean, and why will these tool combinations "lead to decreased performance"? Providing some simple examples can make it easier to read.

[1] Zhao, Andrew, et al. "Expel: Llm agents are experiential learners." Proceedings of the AAAI Conference on Artificial Intelligence. Vol. 38. No. 17. 2024.

[2] Fu, Yao, et al. "Autoguide: Automated generation and selection of state-aware guidelines for large language model agents." arXiv preprint arXiv:2403.08978 (2024).

[3] Chen, Zhe, et al. "Internvl: Scaling up vision foundation models and aligning for generic visual-linguistic tasks." Proceedings of the IEEE/CVF Conference on Computer Vision and Pattern Recognition. 2024.

[4] Wang, Wenhui, et al. "Image as a foreign language: Beit pretraining for vision and vision-language tasks." Proceedings of the IEEE/CVF Conference on Computer Vision and Pattern Recognition. 2023.

**Questions:**

In line 160, "the comparator samples a group of data samples (question-answer pairs), executes the current actions for each question," but how can the comparator execute the current actions? What are the current actions for the comparator?

The text in Figure 3 is too small to read and the meaning of the marked text is difficult to understand.

**Limitations:**

There is no separate Limitation or Broader Impacts section in the paper. In addition, the authors said, "We did an extensive survey on related work in the area of LLM agents, agent optimization, LLM agent for retrieval, and further discuss their limitations," but the discussion of the limitations in this paper is required.

---

> ### Author Rebuttal · Authors · 2024-08-05
>
> We appreciate your comments! We carefully justify our novelty and clarify the misunderstandings. We will be grateful for your patience of reading our response:
>
> ---
>
> ## **Comment 1: Comparison with Expel [1] and Autoguide [2]**
> We apologize for missing these important and relevant papers. Thank you for helping us do right by the existing work. While agree that they are relevant, we summarize the key differences:
>
> ||ExpeL|AutoGuide|AvaTaR|
> |:--|:--|:--|:--|
> |**Contrast Over**| Successful and failed trajectories (action sequences)|Trajectories|Positive & negative data samples|
> |**Training Phase**|Two phases: sample experiences, then extract insights|Two phases, similar to ExpeL|One phase training with iterative instruction generation and action refinement|
> |**Inference Method**|In-context generation with extracted insights|Iterative generation with state guidelines|**Direct generalization with optimized action sequence|
>
> ---
>
> Specifically, ExpeL is an innovative method that leverages experiences across multiple training tasks, and AutoGuide introduces state-aware guidelines for agents. All three methods learn from past experience.
>
> However, the high-level ideas are different. Specifically, AvaTaR **contrasts positive and negative queries for the action sequence generated by the actor at current step**, while both ExpeL and AutoGuide **contrast failed and success action sequences given an instance**. As an analogy with RL, the instructions from AvaTaR comparator is **on-policy** (target at current action sequence), while ExpeL and AutoGuide requires **off-policy** approximation with data pool. Moreover, AvaTaR conducts end-to-end training and direct inference on up to 4k testing queries in total, which achieves higher scalability compared to in-context inference methods.
>
> We added discussions in our revision. Once again thank you very much for helping us position our work better.
>
> ---
>
> ## **Comment 2.1: “The paper claims their method achieves SOTA on the benchmarks”**
> Our statement in the abstract - “We find AvaTaR consistently outperforms SOTA  approaches” - refers to SOTA in the scope of LLM agents. We apologize for the confusion and have clarified the statement.
>
> ---
>
> ## **Comment 2.2: “they only compared with relatively basic methods”**
>
> - ### **a) Recap: Baselines used**
>   - For STaRK, we compared **all available methods** reported by the benchmark and two agent methods: ReAct and Reflexion.
>   - While image retrieval tasks are less explored by previous agent works including Expel and Autoguide, we made **non-trivial adaptations** to apply ReAct and Reflexion.
>
> - ### **b) Justification: Are the baselines sufficient?**
>   We believe AvaTaR is compared against essential baselines. For reference, Expel compared with ReAct and Act. While ReAct generally outperforms Act, we use Reflexion as an alternative. Similarly, Retroformer [3] used ReAct and Reflexion as baselines.
>
> ---
>
> ## **Comment 2.3: “outdated embedding-based retrievers ... dominated by the VLMs (Internvl and Beit)”**
> - ### **a) Recap: What are the embedding-based retrievers we used?**
>   - For STaRK, we used `text-embedding-ada-002`, a competitive model on the Mteb leaderboard during the time of this work.
>   - For Flickr30k-entities, we used `clip-vit-large-patch14`, a commonly used VLM.
> - ### **b) Clarification: The embedding models are not only baselines, but also tools for agents**
>   Table 5 and 6 listed a set of tools for **all agent methods**. E.g., `GetClipImageEmbedding` uses the same CLIP model as the baselines for fair evaluation.With the **given tools**, our goal is to improve agents' tool-use ability, which leads to improved performance.
> -  ### **c) Justification: AvaTaR v.s. VLMs or AvaTaR+VLMs?**
>    In the above setting, AvaTaR achieve proof-of-concept results which outperforms the baselines **with the same tools or embedding model**.
>
>    Differently, `Internvl (6b)` and `Beit (1.6b)` pretrain large VLMs, which are not fairly comparable with AvaTaR since it used a much smaller clip model with 427m parameters. More importantly, *the goal of AvaTaR (improving agents) and the pretraining methods (training powerful VLMs) are different*.
>
>     In fact, **these two method classes are not conflicting. Both VLMs can be tools leveraged by AvaTaR!** This inspires a practice future direction to construct powerful tools from open world.
>
> ---
>
> ## **Comment 3: Missing comparison on FLICKR30K-ENTITIES**
> The results include all available methods. Compared with Table 2, Dense Retriever is a finetuned text encoder from STaRK; QAGNN is for knowledge graphs; Multi-VSS chunks documents and is not applicable to images. We added explanations for clarity.
>
> ---
>
> ## **Comment 4: Statements in L152-154**
>  - **“Per-sample instructions”**: Instructions generated for a failure/negative query instance.
> - **Example**: When answering query `What are some recommended traditional setup fishing rigs from the Thill brand?` Reflexion tend to rely on specific details like "Thill". E.g., It (1) computes a token match score, and if there is a match (2) returns all fishing rigs items, which fails to generalize when user requests other Thill items"
>
> ---
>
> ## **Comment 5: Limitations**
> We added a limitation section. Due to character limit, please refer to our response to `reviewer CkbF`. Thanks!
>
> ---
>
> **Question 1: Clarification of comparator**
> - “Current actions” refers to the action sequence generated by the actor in the current training step.
> - “Execute the current actions” means evaluating the current action sequence on sampled queries. Actions are executed by actor, we clarify it in the revision.
>
> **Question 2: Figure 3.**
> The light orange and blue highlight features of positive and negative queries, respectively. We will adjust the caption size for readability.
>
> ---
>
> # **Summary**
> We sincerely hope we address your concerns. We would very much appreciate it if you could reconsider your evaluation if some concerns are addressed. Thank you very much!

---

> ### Comment · Reviewer_JqLP · 2024-08-10
>
> Thanks for the detailed responses, but they have not fully addressed my concerns.
>
> 1- Even though there are differences in details, the existence of papers like ExpeL and AutoGuide clearly reduces this paper's innovation in the direction of self-optimization of LLM agents. Given ExpeL and AVATAR both utilize the comparator to optimize the prompt, the main innovation of this paper would be replacing 'off-policy' with 'on-policy' learning. However, as 'off-policy' and 'on-policy' are both useful for RL, the authors should perform more experiments and studies to show that their 'on-policy' methods are better than the 'off-policy' version.
>
> 2- Previous works (ExpeL and Retroformer were published in Aug 2023) used ReAct and Reflexion as baselines, which cannot be the reason that this paper still uses the same baseline. The field of LLM Agents is a rapidly developing field. In the year after reflexion (published in Mar 2023), a large number of high-quality methods have been proposed, e.g., [1][2][3]. However, in the experiment, the author ignored the comparison with these cutting-edge methods. In addition, in the introduction and related work of the paper, the author repeatedly pointed out that the previous LLM Agent methods via self-optimizing cannot solve complex problems. However, there is no comparison with these methods in the experiment to verify these claims.
>
> Therefore, the authors should keep up with recent methods, and consider these approaches in their experiments, rather than just the basic methods.
>
> 3- The method proposed in this paper is not only applicable to retrieval tasks but also to general agent tasks. Therefore, it should also be experimented on some common and difficult benchmarks. As I said before, one of my concerns is that STARK is a very new benchmark (published in April 2024), and Flickr30k-entities is a very rarely used benchmark for LLM Agent. Achieving good results on LLM agent benchmarks that have been studied more, such as HotpotQA [4], AgentBench [5], and WebArena [6], the method can be more convincing.
>
> 4- For the question about the clarification of the comparator, your answer should be added to the revised version because the presentation of the original text is unclear and misleading.
>
> 5- One more new question: I found the authors implemented AVATAR with a batch size of 20: "the metric Recall@20 for constructing positive and negative queries, and hyperparameters (l = h = 0.5, b = 20)". Given that 20 trajectories of Actors can be quite long, can the authors provide the context length required by the comparator and the token overhead required for AVATAR optimization?
>
> [1] Zhu, Zhaocheng et al. “Large Language Models can Learn Rules.” ArXiv abs/2310.07064 (2023): n. pag.
>
> [2] Majumder, Bodhisattwa Prasad et al. “CLIN: A Continually Learning Language Agent for Rapid Task Adaptation and Generalization.” ArXiv abs/2310.10134 (2023): n. pag.
>
> [3] Qian, Cheng et al. “Investigate-Consolidate-Exploit: A General Strategy for Inter-Task Agent Self-Evolution.” ArXiv abs/2401.13996 (2024): n. pag.
>
> [4] Yang, Zhilin et al. “HotpotQA: A Dataset for Diverse, Explainable Multi-hop Question Answering.” Conference on Empirical Methods in Natural Language Processing (2018).
>
> [5] Liu, Xiao et al. “AgentBench: Evaluating LLMs as Agents.” ArXiv abs/2308.03688 (2023): n. pag.
>
> [6] Zhou, Shuyan et al. “WebArena: A Realistic Web Environment for Building Autonomous Agents.” ArXiv abs/2307.13854 (2023): n. pag.

---

> ### Author Response · Authors · 2024-08-13
> **2nd Batch response**
>
> We appreciate the reviewer for further communication. We hope our response this time could mostly address your concerns.
>
> ### **TL;DR;**
> - We clarify Avatar's novelty, especially on AvaTaR $\neq$ ExpeL + on-policy training.
> - We show that AvaTaR outperforms ExpeL and Retroformer on HotpotQA. We also show the performance and scalability advantage of AvaTaR over ExpeL on STaRK-MAG;
> - We show that AvaTaR works well on new QA datasets (HotpotQA, ToolQA, ArxivQA)
>
> ---
>
> Before started, we make the following terms consistent to avoid confusion:
> - Queries = instances (used in AvaTaR) = tasks (used in ExpeL)
> - Actions to answer queries = action sequences (AvaTaR) = trajectories (ExpeL)
>
> ---
>
> **Comment 1: About novelty**
>
> While we pointed out "on-policy" and "off-policy" as one difference between AvaTaR and ExpeL, the novelty of AvaTaR goes beyond that. Here are our reasons:
>
> - **What to contrast**: This is relevant to the reviewer's question about batch size - *"Given that 20 trajectories of Actors can be quite long."* We believe this is a misunderstanding (please let us know if otherwise). Please note that AvaTaR contrasts positive and negative queries, unlike ExpeL, which contrasts trajectories. Therefore, the context length is less of an issue since the queries are mostly short, we will provide detailed statistics.
> - **Benefits of contrast queries**:
>   Following the last point, the reviewer actually pointed out one benefit of contrasting queries - it allows us to contrast among a batch of pos/neg queries `(batch_size=b)` rather than a well-performing and an under-performing trajectory (`batch_size=2`). This design offers better scalability and generalization ability. We provided intuitions in L173-184, and we repeat them here for your convenience:
>
>
>    ```
>    Moreover, as contrastive reasoning directly targets disentangling the performance gap related to input patterns and how they are handled differently by the tools, it is particularly effective in helping comparators differentiate and select tools for use. Finally, by identifying systemic flaws across a wide array of negative queries, comparator generates modifications that are not only tailored to individual samples but also to diverse data samples, offering benefits for better generalization to novel samples.
>    ```
>
>   This design enables AvaTaR to generate more holistic instructions from the insights on multiple queries and/or generalize the final action sequence to hundreds of testing queries without test inference, which are validated by the strong generalization in our existing experiments.
>  - **AvaTaR $\neq$ ExpeL + on-policy training**: Due to the differences in design, AvaTaR is able to conduct on-policy training since it directly optimizes the action sequence for multiple queries. However, one can imagine it would be hard for ExpeL to conduct on-policy training with trajectory comparison.
>
> ---
>
> **Comment 2: Comparison with more baselines**
>
> We added ExpeL and Retroformer as our baselines in our paper. We firstly compare them with AvaTaR on HotpotQA on the dev subset (100 queries) in their repositories. Here are the results, where all the baseline results are reported by ExpeL and Retroformer):
>
> ||HotpotQA (EM)|
> |:---|--:|
> |Act|29%|
> |ExpeL|39%|
> |ReAct |40% |
> |Reflexion|46%|
> |Retroformer (#retry=1)|51%|
> |AvaTaR|55%|
>
> For AvaTaR, we take 100 training queries to optimize prompts for 10 steps and set the number of retries as 0, which achieves the best performance. Then, we compare AvaTaR with ExpeL on STaRK-MAG. STaRK datasets involve over 20 tools, leading to long action sequences and high token overhead for ExpeL to process and evaluate. Due to this, we randomly sampled 100 training queries and evaluated ExpeL on 50 testing queries, comparing with the other methods on the same set.
>
>
> |||MAG|(#Test=50)||
> |:---|:---|:---|:---|:---|
> ||Hit@1|Hit@5|Recall@20|MRR|
> |Dense Retriever|16.00|40.00|51.84|27.39|
> |QAGNN|20.00|52.00|49.71|36.39|
> |VSS|40.00 |58.00|55.93|47.76|
> |Multi-VSS|32.00|58.00|58.81|43.58|
> |ReAct|46.00|60.00|54.67|50.92|
> |ExpeL|40.00|58.00|55.94|47.43|
> |Reflexion|48.00| 64.00| 57.43|52.31|
> |AvaTaR-C|44.00|60.00|52.49|50.16|
> |AvaTaR|52.00| 64.00|53.86 |56.74|
>
> For ExpeL, we used `text-embedding-3-large` embedding model to retrieve insights. We found ExpeL’s performance similar to ReAct, which might be because the STaRK queries are diverse, therefore require more training data to gather enough experience.
>
> We will have ExpeL results on the other two STaRK datasets and include a table similar to Table 2 in our paper. Hopefully, these will address your concern about the comparison with previous LLM Agent methods and justify our claims.

---

> ### Author Response · Authors · 2024-08-13
> **2nd Batch response (Continue)**
>
> **Comment 3: More benchmarks.** Our experiments on HotpotQA improve this aspect. Please also see our response to `Reviewer CkbF` for AvaTaR results on ArxivQA and ToolQA.
>
> **Comment 4: Clarity on comparator.** Thanks! Yes, we made sure the statement is clear now.
>
> **Comment 5: Content requirements and token overhead.** On STaRK-MAG, the context length for the comparator is approximately 4k tokens per step, including initial instructions and tool usage (with pos/neg queries taking around 0.8k). The actor's token cost, including memories, is around 8k. Running AvaTaR for 50 steps accumulates about 600k tokens, costing under $10 with gpt-4-turbo. There is no inference cost on STaRK dataset as we directly apply the action sequence.
>
> As a reference, we also compute the token cost for ExpeL on STaRK-MAG: Training token cost: 357k in total (3.57k per task); Testing token cost: 389k in total (7.8k per task).
>
> In this comparison, we believe AvaTaR has an advantage in scaling especially when the number of testing queries are large.
>
> For HotpotQA, the comparator's context length is around 2k tokens per step (with pos/neg queries at 0.5k). The actor's token cost ranges from 2k to 8k, depending on the number of actions. Running 10 steps accumulates around 80k tokens.
>
> *Additional related works.** Thanks! We have also added them to our related work.

---

### Official Review · Reviewer_CkbF · 2024-07-11

**Soundness:** 3
**Presentation:** 3
**Contribution:** 3
**Rating:** 6
**Confidence:** 4

**Summary:**

This paper introduces AVATAR, a novel framework for optimizing large language model (LLM) agents to effectively use provided tools and improve performance on complex multi-step tasks, with a focus on retrieval tasks. The key innovation is a comparator module that generates holistic instructions to improve the actor (main agent) through contrastive reasoning on batches of well-performing and poorly-performing queries.
The authors demonstrate AVATAR's effectiveness on four challenging retrieval datasets from the STARK and FLICKR30K-ENTITIES benchmarks. Results show significant improvements over state-of-the-art baselines like ReAct and Reflexion.

**Strengths:**

Novel approach: The comparator module using contrastive reasoning on query batches is an innovative way to generate holistic instructions for improving agent performance. This addresses limitations of per-sample instruction approaches.

Clear motivation and explanation: The paper draws insightful analogies to concepts like batched training and gradient computation in neural networks to explain the intuition behind the approach. This makes the core ideas easy to understand.

Strong empirical results: AVATAR consistently outperforms strong baselines across multiple datasets, with significant improvements on key metrics. The ablation study clearly demonstrates the value of the comparator module.

**Weaknesses:**

Limited scope of evaluation: While the retrieval tasks are complex, it would be valuable to see AVATAR applied to a broader range of tool-use benchmarks to demonstrate generality.

Lack of comparison to finetuning: The paper does not discuss or compare to alternative approaches like directly finetuning the actor model via rejection sampling. It's unclear how AVATAR compares to such methods in terms of performance and efficiency.

Scalability considerations: The paper does not thoroughly address how the method scales with increasing numbers of tools or more complex task structures.

**Questions:**

Regarding the use of the memory bank as in-context learning examples,
How does AVATAR compare to directly finetuning the actor model in terms of performance, efficiency, and data requirements? What are the key advantages of the instruction-based approach?
How does the method handle scenarios where the action model improves significantly, making it difficult to find negative examples for the contrastive learning process? Is there a strategy for addressing this?
Have you explored applying AVATAR to other types of tool-use tasks beyond retrieval? What challenges do you anticipate in extending to more diverse task types?

**Limitations:**

The authors have not adequately addressed the limitations of their work. I recommend adding a dedicated limitations and broader impacts section to address these points more thoroughly.

This should include: Discussion of computational requirements and scalability limitations, potential failure modes or scenarios where the method may struggle

---

> ### Author Rebuttal · Authors · 2024-08-05
>
> We are grateful for your positive feedback! We provide point-to-point responses:
>
> ---
>
> ## **Comment 1: Applying AvaTaR to more tool-use benchmarks**
>
> Thanks! We've conducted initial experiments on ArxivQA and are currently running tests on ToolQA. Please stay tuned! For ArxivQA, we randomly sampled 500/200 training/testing queries, providing three web search tools and five LLM tools such as summarization.
>
> | | CoT| ReAct | AvaTaR |
> |:---|:---|:--|:--|
> |Correctness by LLM Judge|$58.0$\%|$73.5$\%|$85.5$\%|
>
> We are happy to provide more details. Notably, AvaTaR achieved a substantial improvement (12%) over ReAct. For our other evaluation datasets, we highlight that they are more comprehensive than some existing tool-use datasets. For example, ToolEye [1] offers a fine-grained system but only has 54 retrieval queries, while MetaTool [2] involves only one or two tools per query. In contrast, our datasets involve action sequences with over ten tools (Figure 2) and span multiple modalities, totaling over 4k testing queries.
>
> ---
>
> ## **Comment 2 & Question 1,2: Comparison between finetuning methods and AvaTaR (with memory module)**
>
> Thanks for the insights! We agree that finetuning is a potential way to improve the actor LLM. We give an overview first and then conduct the comparison.
>
> - **Overview**: Several benchmarks and methods focus on fine-tuning tool-augmented LLMs. ToolBench (ICLR’24) [3] and GPT4Tools [4] create instruction-tuning datasets to enhance tool-use capabilities. Similarly, API-Bank [5] provides a dataset to improve planning, retrieval, and tool-calling skills. ToolAlpaca [6] generates tool-use corpus to develop generalized tool-use abilities in smaller LMs through fine-tuning.
>
> However, we believe **finetuning approaches may be challenging to adapt to our tasks** and AvaTaR has better advantages in the following aspects:
>
> - **Data Requirements**: Previous works [4,5,6,7] require large-scale datasets involving thousands of tools for LLM finetuning, while some tasks use fewer than a hundred tools, making data generation challenging. Moreover, creating instructions with ground truth tool use from ChatGPT [3,4] or multi-agent frameworks [5,6] can be difficult, especially for retrieval tasks with external knowledge bases. In contrast, AvaTaR needs only a subset of training QA and tool descriptions, making its data requirements less demanding.
>
> - **Efficiency:**
>   - **Inference:** AvaTaR applies a fixed solution to testing queries without additional memory modules or comparators, making its inference efficiency comparable to or better than finetuning methods.
>   - **Training:** AvaTaR reduces human effort by eliminating the need to design GPT prompts or multi-agent frameworks for annotating tool usage. For time cost, AvaTaR is observed to be efficient (c.f. Figure 4), typically converges within 50 iterations. At each iteration, we track only the top-5 action sequences with the best performance, minimizing token costs from the memory bank.
>
> - **Performance:** The reasoning ability of the actor LLM impacts task performance, motivating the use of state-of-the-art LLMs like Claude3, which are closed-source and mostly unavailable for finetuning. Besides more flexibility to use these models, AvaTaR can use any finetuned model with enhanced tool usage abilities as the actor LLM, which can be further trained by the comparator.
>
> We added this discussion in our revision. Thank you for highlighting finetuning methods to help us better position our work.
>
> ---
>
> ## **Question 3: Strategy for selecting negative samples**
>
> Great question! This inspires us to implement an adaptive threshold strategy. Specifically, we track evaluation results on a subset of training queries from the past several epochs and use the median performance as the threshold for the current epoch. This helps ensure a sufficient number of negative samples for selection.
>
> ---
>
> ## **Question 4 and Comment 3.2: Challenges of extending AvaTaR**
>
> We have extended AvaTaR to general QA tasks with newly added datasets. With recent support for AvaTaR in the DSPy [7] library, we expect its application to more tasks, such as coding problems, where positive/negative sampling can be determined by unit test results.
>
> A key challenge for complex tasks, like visual reasoning [8,9], is that the input space can make extracting insights difficult. These tasks often require stronger reasoning to identify patterns compared to simpler formats like natural language queries. We added this to the future work.
>
> ---
>
> ## **Comment 3.1: Scalability and limitations**
> Our paper used around 25 tools per dataset, offering a decent level of complexity. Further, we discuss the limitations:
> - **Scalability**: With LLM context lengths increasing (up to 128k), AvaTaR can scale to handle hundreds of tools and complex tasks, but practical limitations like increased latency may impact performance. Future efforts could integrate finetuned tool-augmented LLMs as actors or comparators to facilitate smooth scaling.
> - **Computation Requirements**: Scaling AvaTaR can increase computational costs due to the need to manage longer contexts and multiple tool interactions, leading to higher expenses.
> - **Potential Failure Modes**: While AvaTaR generalizes well to testing queries, performance may degrade if queries require novel tool combinations that AvaTaR hasn't trained for. Robust monitoring and adaptive learning strategies may help mitigate these risks.
>
> We added a limitation section with extended discussions.
>
> ---
>
> # **Summary**
>
> We appreciate your approval on our novelty, motivation, and effectiveness. We hope we address your concerns with 1) more experiments on two QA datasets (ongoing), 2) extensive comparison with finetuning methods, 3) a limitation section. We are more than happy to follow up.
>
> We also kindly ask if you could reevaluate our work if some concerns are addressed. Thank you for your insights and support again!
>
> *Reference is in the uploaded PDF*

---

> ### Author Response · Authors · 2024-08-12
> **(Continue) Experiments on more tool-use benchmarks**
>
> Hi Reviwer CkbF,
>
> Thanks for waiting for the follow-up results on ToolQA! We introduce the setup and present the full results here:
>
> - **Datasets**: SciREX (a dataset for document-level information extraction based on full-length ML scientific papers) and Agenda (personal agenda questions based on a private knowledge base) are the two (and only two) datasets based on external text corpora from ToolQA. Each dataset has `easy` and `hard` splits.
> - **Configuration for AvaTaR:** We randomly split the questions by 40%:60% for training and testing, as official dataset splits are not provided. We set the maximum epoch to 5.
> - **Tools:** We use the knowledge base tools provided by ToolQA and web search tools for all agent methods.
> - **Agent Backbone:** We use `gpt-4o` for the LLMs of AvaTaR and the baselines, which are evaluated on our testing split for fair comparison.
>
> |            | SciREX | SciREX | Agenda | Agenda |
> |------------|--------|--------|--------|--------|
> |            | Easy   | Hard   | Easy   | Hard   |
> | CoT        | $0.0$%   | $0.0$%   | $0.0$%   | $0.0$%   |
> | ReAct      | $8.3$%   | $18.3$%  | $31.6$%  | $11.7$%  |
> | Reflexion  | $10.0$%  | $13.3$%  | $30.0$%  | $13.3$%  |
> | AvaTaR     | $11.6$%  | $21.7$%  | $36.7$%  | $23.3$%  |
>
> We observed consistent improvements across these two datasets, with a significant improvement on the hard queries of Agenda.
>
> We hope the experiments on ToolQA and ArxivQA can further validate the effectiveness and applicability of AvaTaR on more tool-use datasets. We are happy to provide any details (e.g., insights or example instructions) that are not covered here for conciseness.

---

### Official Review · Reviewer_vMWD · 2024-07-12

**Soundness:** 3
**Presentation:** 3
**Contribution:** 3
**Rating:** 6
**Confidence:** 3

**Summary:**

This paper proposes a new framework AVATAR for LLM agent that operates in two stages:
-The first stage is optimization during the training process, which integrates the LLM comparator component into the AVATAR. The comparator summarizes holistic prompts from positive and negative queries and iteratively optimizes LLM actor using the prompts.
-The second stage is deployment during the inference process, where participants respond without the involvement of the comparator.

**Strengths:**

Developing prompts for LLM agents is heuristic and laborious, but this paper proposes a new method called AVATAR, which automatically iteratively generates holistic prompts from positive and negative queries. Experimental results show that AVATAR outperforms SOTA methods.
and some points summary as following:
1. It is a novel framework, comparing with the current SOTA approach such as ReAct, the Comparator component design is reasonable that it generates prompts from positive and negative queries.
2. It is new SOTA of LLM Agent. Experiments are conducted against current SOTA approaches as baseline, the framework outperform the current SOTA
3. It is a common framework, and it is effective with both Claude and GPT4.

**Weaknesses:**

1. The optimization is data-driven, so comparator works effectively when the positive and negative queries are well sampled and balanced during iteration; when to stop iteration and how to sample the queries; it is not clearly clarified.
2. Regarding the construction of positive and negative queries, it is necessary to conduct experiments on the lower and upper bounds(l&h) to select appropriate values for optimal performance.
3. Regarding the memory bank, it is a important component to swap in and out according to data quality or balanced distribution,
it is not well designed and discussed.

**Questions:**

Is "AIR (gpt4)" in Table 3 a typo?

---

> ### Author Rebuttal · Authors · 2024-08-05
>
> We sincerely appreciate your time and insightful comments! Here are our point-to-point responses:
>
> ---
>
> ## **Comment 1: Clarification about when to stop iteration and how to sample the queries**
>
> Thanks! We apologize for the brief information there. We added the following details in our revision:
>
> - **Early stopping for iteration control**: In our implementation, we train AvaTaR for a fixed number of epochs and select the action sequence with the highest validation performance. Similar to training deep neural networks, early stopping is efficient when performance on a hold-out subset doesn’t improve for a set number of epochs. This approach can further save time and cost for our framework.
>
> => ***We included more details in the experiment section and made early stopping available in our implementation.***
>
> - **Positive & Negative Query Sampling**: We employed random sampling, a simple yet effective method. Meanwhile, to ensure a balanced set of positive and negative queries (totaling $b$ queries) for contrastive reasoning, our training framework uses a larger sampling batch size, such as $1.5b$. The sampled queries are classified as positive or negative based on performance and then balanced. For example, if there are $0.5b$ positive and $b$ negative queries in the **sampling batch**, we further randomly sample $0.5b$ from the negative queries, resulting in a **training batch** of $0.5b$ positive and $0.5b$ negative queries.
>
> => ***We added the detail to Section 4.2, Step 1: Constructing positive and negative queries.***
>
> We hope the above discussion can address your concern about the clarity of our work!
>
> ---
>
> ## **Comment 2: Hyperparameter search on  lower and upper bounds ($\ell / h$)**
>
> Good question! Yes it is important to search these two hyperparameters.
>
> In our previous experiments, we found these two hyperparameters are pretty robust *w.r.t.* small adjustments. Therefore, we set $\ell=h=0.5$ for all datasets. But now we draw more clear conclusions with additional experiments. To recap, queries where the Recall@20 metric is higher than $\ell$ are grouped into positive and queries where the Recall@20 metric is lower than $h$ are grouped into negative groups. Thus we have $\ell \geq h$.
>
> Specifically, we tested multiple combinations of $\ell$ and $h$. We conducted experiments on STaRK-Amazon only due to the time and computational cost involved. And here are the Hit@1 results:
>
> |  | $h=0.3$ | $h=0.4$ | $h=0.5$ |
> |:---:|:---:|:---:|:---:|
> | $\ell=0.5$  | $48.32$ | $50.01$ | $49.87$ |
> | $\ell=0.6$  | $47.89$ | $49.56$ | $50.45$ |
> | $\ell=0.7$  | $47.75$ | $48.56$ | $49.34$ |
>
> Interestingly, we obtained **a better Hit@1 result** than the one reported in the paper when $\ell = 0.6$ and $h = 0.5$. Results for other metrics can be found in the uploaded PDF.
>
> **Key Observations:**
>
> - The performance is **robust** to the large adjustments, with variations of 2.7% in Hit@1.
> - We observe some **decline when the gap between $\ell$ and $h$ is too large**. This is likely due to omitting part of the training queries (the performance on which are within $(h, \ell)$.
> - We see **slight improvement when there is a moderate gap** between $\ell$ and $h$, which could help establish a clearer pattern difference between positive and negative queries without “sacrificing” too many training queries.
>
> We add the above results and discussion to Appendix B. We hope our study can address your concern about hyperparameter selection. In general, our framework relies on a very small number of hyperparameters, *i.e.*, $\ell$, $h$, batch size $b$, and training epochs. We believe the study on $\ell$ and $h$ demonstrate the robustness of our framework.
>
> ---
>
> ## **Comment 3: Design of memory bank and its influence on the optimization process**
>
> Thanks a lot for this insightful comment!
>
> For memory bank construction, we follow Reflexion [1], which also exhibits long-term memory. Alternatively, our framework allows for the direct plug-in of other kinds of memory banks. For example, a dynamic and multi-layered memory structure proposed by [2] can be used for storing knowledge and experience dynamically from past training. We believe a more advanced memory component is a great addition to our optimization framework, which has been less explored in previous studies.
>
> We also want to point out that our main contribution is orthogonal to the design of memory banks. Specifically, our novelty and success largely attribute to the comparator in the optimization module, which is validated in the ablation study between AvaTaR and AvaTaR-C.
>
> Following your suggestions, in our revision, 1) we added more discussion about the memory bank, and 2) we pointed out that enhancing the memory bank could be a key innovation in the future. Note that while revision is not allowed during rebuttal, we are happy to copy the updated paragraphs here if you would like to see them!
>
> We hope the memory module is well-discussed now. We thank you for these insightful suggestions again, which inspired us for future extensions.
>
> **Typo**: Good catch! Yes it should be "AvaTaR," and we've fixed it in the revision. Thanks!
>
> ---
>
> # **Summary**
>
> We thank you for your time and insights! We hope our 1) clarification on the training process, 2) experiments on hyperparameters, and 3) discussions on memory banks address your concerns well.
>
> At last, we would greatly appreciate your support and reconsideration given our response.  We would like to emphasize that our main contribution is the development of an optimization framework featuring a novel comparator module. This module enhances contrastive reasoning and improves generalization, allowing our approach to significantly outperform baseline methods. Thank you in advance!
>
> ---
>
> **Reference**
>
> [1] Shinn et al. 2023. Reflexion: language agents with verbal reinforcement learning. In NeurIPS.
>
> [2] Zhong et al. MemoryBank: Enhancing Large Language Models with Long-Term Memory.

---

> > ### Comment · Reviewer_vMWD · 2024-08-13
> > **Thanks for your response, no more comments**
> >
> > See it above.

---

### Author Rebuttal · Authors · 2024-08-05

# **General response**

We truly appreciate the reviewers' efforts and valuable suggestions in reviewing our paper. We are glad that all/most reviewers reached a positive consensus on our work's presentation, motivation, novelty, and experimental effectiveness. Here is a summary of the reviewers’ major feedback and our corresponding actions:

-----

| | Reviewer JqLP | Reviewer CkbF | Reviewer vMWD | Action/Summary |
|:---|:---|:---|:---|:---|
| **Presentation** | “Presentation: good” | “Clear motivation and explanation” | “demonstrate the proposed method well” | `NA`|
| **Novelty** | “novel framework, …, the Comparator component design is reasonable” | “The comparator module … is an innovative way” | “the method…has no innovation compared with Expel [1] and Autoguide [2]” | RE reviewer vMWD: `We carefully read the related works shared by reviewer vMWD. While highly relevant, we  found essential differences between the mentioned works and AvaTaR in terms of novelty and insights. Please see our response to reviewer vMWD for a detailed explanation.` |
| **Empirical Improvements** | “effective with both Claude and GPT4” | “AvaTaR consistently outperforms strong baselines across multiple datasets, with significant improvements on key metrics” | “The experiments are performed … to show the effectiveness of the methods.” | `NA`|
| **Evaluation Setting** | NA| Suggestions on a broader application of AvaTaR on tool-use benchmarks | “The paper.. (1) only compared with relatively basic methods, … (2) employs several outdated embedding-based retrievers…(underperform) Internvl and Beit VLMs. ” | RE reviewer CkbF: `We add additional experiments on two QA and tool-use benchmarks - ArxivQA and ToolQA (ongoing).` RE reviewer vMWD: (1) `We justified the baselines included, where ReAct and Reflexion are two prevailing agent methods.` (2) `We emphasize that our goal is to optimize agents and improve their tool-use abilities. We use clip-vit-large-patch14 (427 million parameters) embedding model as a tool for all of the agent methods. In contrast, Internvl (6b parameters) and Beit (1.6b parameters) study pretraining methods for large-scale VLMs, which have different goals and settings than AvaTaR.` |
| **More Study and Discussion on AvaTaR** | Pos & neg query selection and the contribution of memory bank | (1) Scaling up with more tools and complex task structures. (2) Comparison between AvaTaR and finetuning methods | NA| RE reviewer JqLP: `We added experimental studies on pos & neg query selection.` RE reviewer CkbF: (1) `We highlight the level of complexity that our tasks reached and discuss future directions on scaling up.` (2) `We added extensive discussion between AvaTaR and finetuning methods  in terms of performance, efficiency, and data requirements` |


Moreover, following the suggestions from Reviewers CkbF and vMWD, we added a dedicated `limitation section` (see response to the reviewers) in our final draft.

------

# **Summary**

We thank the reviewers for their suggestions which make our work more solid. We have improved our manuscript accordingly. We hope our responses can clarify any confusion and alleviate the remaining concerns above.

We would be thrilled if you could let us know whether your concerns have been addressed or if you have any follow-up questions!

— Best,

  Authors of Paper 5514

---

> ### Author Response · Authors · 2024-08-14
>
> Correction: In the summary table above, the reviewer labels 'JqLP' and 'vMWD' should be switched, which matches the sequence order on the webpage. Apologies.

---

### Decision · Program_Chairs · 2024-09-25

**Decision:**

Accept (poster)

**Comment:**

The paper initially received 2 Weak Accepts and 1 Reject ratings. While reviewers praised the presentation, effectiveness and interesting ideas, there were concerns on lack of details, lack of novelty and missing experimental comparisons. The rebuttal and discussions helped significantly and assuaged most concerns, leading to substantial discussion between the reviewers and resulting in a mostly positive score. The AC believes that the paper has merit and can be published at NeurIPS. We require that the clarifications, rewritings and additional information provided in the rebuttal be incorporated in the camera-ready version.